# Continual Learning with Evolving Class Ontologies

**Zhiqiu Lin**[1]   **Deepak Pathak**[1]   **Yu-Xiong Wang**[2]   **Deva Ramanan**[1,3]*   **Shu Kong**[4]*

[1]CMU      [2]UIUC      [3]Argo AI      [4]Texas A&M University

Open-source code in webpage

## Abstract

Lifelong learners must recognize concept vocabularies that evolve over time. A common yet underexplored scenario is learning with class labels that continually refine/expand old classes. For example, humans learn to recognize `dog` before dog breeds. In practical settings, dataset *versioning* often introduces refinement to ontologies, such as autonomous vehicle benchmarks that refine a previous `vehicle` class into `school-bus` as autonomous operations expand to new cities. This paper formalizes a protocol for studying the problem of *Learning with Evolving Class Ontology* (LECO). LECO requires learning classifiers in distinct time periods (TPs); each TP introduces a new ontology of "fine" labels that refines old ontologies of "coarse" labels (e.g., dog breeds that refine the previous `dog`). LECO explores such questions as whether to annotate new data or relabel the old, how to exploit coarse labels, and whether to finetune the previous TP's model or train from scratch. To answer these questions, we leverage insights from related problems such as class-incremental learning. We validate them under the LECO protocol through the lens of image classification (on CIFAR and iNaturalist) and semantic segmentation (on Mapillary). Extensive experiments lead to some surprising conclusions; while the current status quo in the field is to relabel existing datasets with new class ontologies (such as COCO-to-LVIS or Mapillary1.2-to-2.0), LECO demonstrates that a far better strategy is to annotate *new* data with the new ontology. However, this produces an aggregate dataset with inconsistent old-vs-new labels, complicating learning. To address this challenge, we adopt methods from semi-supervised and partial-label learning. We demonstrate that such strategies can surprisingly be made near-optimal, in the sense of approaching an "oracle" that learns on the aggregate dataset exhaustively labeled with the newest ontology.

## 1   Introduction

Humans, as lifelong learners, learn to recognize the ontology of concepts which is being refined and expanded over time. For example, we learn to recognize `dog` and then dog breeds (refining the class `dog`). The class ontology often evolves from coarse to fine concepts in the real open world and can sometimes introduce new classes. We call this *Learning with Evolving Class Ontology* (LECO).

**Motivation**. LECO is common when building machine-learning systems in practice. One often trains and maintains machine-learned models with class labels (or a class ontology) that are refined over time periods (TPs) (Fig. 1). Such ontology evolution can be caused by new requirements in applications, such as autonomous vehicles that expand operations to new cities. This is well demonstrated in many contemporary large-scale datasets that release updated versions by refining and expanding classes, such as Mapillary [52, 55], Argoverse [12, 78], KITTI [4, 23] and iNaturalist [33–37, 69, 75]. For example, Mapillary V1.2 [55] defined the `road` class, and Mapillary V2.0 [52] refined it with fine

---

*Authors share senior authorship.

36th Conference on Neural Information Processing Systems (NeurIPS 2022).

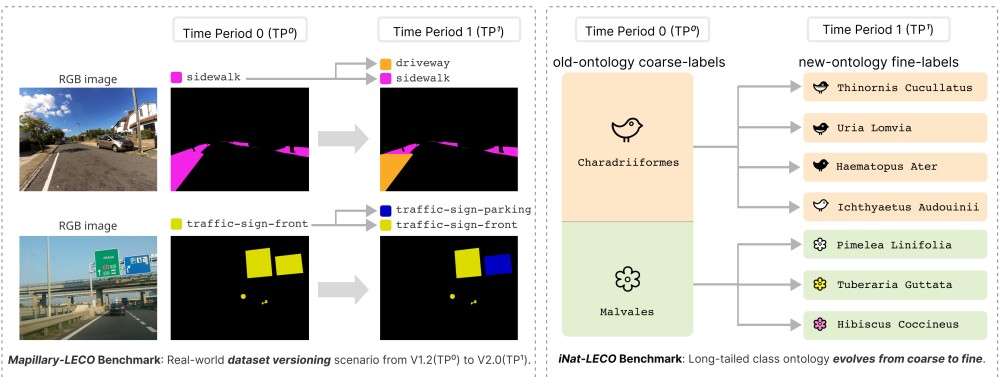

Figure 1: Learning with Evolving Class Ontology (LECO) requires training models in time periods (TPs) with new ontologies that refine/expand the previous ones. **Left:** The Mapillary dataset, which was constructed to study semantic segmentation for autonomous vehicles, has updated versions from V1.2 [55] to V2.0 [52], which refined previous labels with fine-grained ones on the *same* data, e.g., V1.2's `sidewalk` is split to V2.0's `sidewalk` and `driveway`. The visual demonstration with semantic segmentation ground-truth blacks out unrelated classes for clarity. We explore LECO using this dataset. **Right:** To study LECO, we also repurpose the large-scale iNaturalist dataset [33–37, 69, 75] to simulate the coarse-to-fine ontology evolution through the lens of image classification. LECO asks the first basic question: for the new TP that defines new ontology of classes, should one relabel the old data or label new data? Interestingly, both labeling protocols have been used in the community for large-scale datasets. Our study provides the (perhaps obvious) answer – one should always annotate *new* data with the *new* ontology rather than re-annotating the *old*. One reason is that the former produces more labeled data. But this produces an aggregate dataset with inconsistent old-vs-new labels. We make use of insights from semi-supervised learning and learning-with-partial-labels to learn from such heterogenous annotations, approaching the upper bound of learning from an oracle aggregate dataset with all new labels.

labels including `parking-aisle` and `road-shoulder`; Argoverse V1.0 [12] defined the `bus` class, and Argoverse V2.0 [78] refined it with fine classes including `school-bus`.

**Prior art**. LECO requires learning new classes continually in TPs, similar to class-incremental learning (CIL) [18, 47, 74, 85]. They differ in important ways. First, CIL assumes *brand-new* labels that have no relations with the old [47, 49, 60], while LECO's new labels are fine-grained ones that expand the old / "coarse" labels (cf. Fig. 1). Interestingly, CIL can be seen as a special case of LECO by making use of a "catch-all" `background` class [41] (often denoted as `void` or `unlabeled` in existing datasets [14, 22, 49, 52, 55]), which can be *refined* over time to include specific classes that were previously ignored. Second, CIL restricts itself to a small memory buffer for storing historical data to focus on information retention and catastrophic forgetting [10, 47]. However, LECO stores all historical data (since large-scale human annotations typically come at a much higher cost than that of the additional storage) and is concerned with the most-recent classes of interest. Most related to LECO are approaches that explore the relationship of new classes to old ones; Sariyildiz *et al.* [64] study concept similarity but remove common superclasses in their study, while Abdelsalam *et al.* [1] explore CIL with a goal to *discover* inter-class relations on data labeled (as opposed to assuming such relationships can be derived from a given taxonomic hierarchy). To explore LECO, we set up its benchmarking protocol (Section 3) and study extensive methods modified from those of related problems (Sections 4, 5, and 6).

**Technical insights**. LECO studies learning strategies and answers questions as basic as whether one should label new data or relabel the old data[2]. Interestingly, both protocols have been used in the community for large-scale dataset versioning: Argoverse [12, 78] annotates new data, while Mapillary [52, 55] relabels its old data. Our experiments provide a definitive answer – one should always annotate *new* data with the *new* ontology rather than reannotating the *old* data. The simple reason is that the former produces more labeled data. However, this comes at a cost: the aggregate

---

[2]We assume the labeling cost is same for either case. One may think re-annotating old data is cheaper, because one does not need to curate new data and may be able to redesign an annotation interface that exploits the old annotations (e.g., show old `vehicle` labels and just ask for fine-grained make-and-model label). But in practice, in order to prevent carry-over of annotation errors and to simplify the annotation interface, most benchmark curators tend to relabel from scratch, regardless of on old or new data. Examples of relabeling old-data *from scratch* include Mapillary V1.2 [55] → V2.0 [52], and COCO [48] → LVIS [27].

| Benchmark | #TPs | #classes/TP | #train/TP | #test/TP |
|-----------|------|-------------|-----------|----------|
| *CIFAR-LECO* | 2 | 20/100 | 10k | 10k |
| *iNat-LECO* | 2 | 123/810 | 50k | 4k |
| *Mapillary-LECO* | 2 | 66/116 | 2.5k | 2k |
| *iNat-4TP-LECO* | 4 | 123/339/729/810 | 25k | 4k |

Table 1: LECO benchmarks use CIFAR100 [42], iNaturalist [69, 75], and Mapillary [52, 55]. We also vary the number of training data in experiments; as conclusions hold across settings, we report these results in Table **??, ??, ??**.

dataset is now labeled with inconsistent old-vs-new annotations, complicating learning. We show that joint-training [47] on both new and old ontologies, when combined with semi-supervised learning (SSL), can be remarkably effective. Concretely, we generate *pseudo* labels for the new ontology on the old data. But because pseudo labels are noisy [44, 58, 68, 79] and potentially biased [77], we make use of the coarse labels from the old-ontology as "coarse supervision", or the coarse-fine label relationships [26, 45, 61, 73], to reconcile conflicts between the pseudo fine labels and the true coarse labels (Section 6). There is another natural question: should one finetune the previous TP's model or train from scratch? One may think the former has no benefit (because it accesses the same training data) and might suffer from local minima. Suprisingly, we find finetuning actually works much better, echoing curriculum learning [5, 21, 63]. Perhaps most suprisingly, we demonstrate that such strategies are near optimal, approaching an "upperbound" that trains on *all* available data re-annotated with the new ontology.

**Salient results**. To study LECO, we repurpose three datasets: CIFAR100, iNaturlist, and Mapillary (Table 1). The latter two large-scale datasets have long-tailed distribution of class labels; particularly, Mapillary's new ontology does not have strictly surjective mapping to the old, because it relabeled the same data *from scratch* with new ontology. Our results on these datasets lead to consistent technical insights summarized above. We preview the results on the iNaturalist-LECO setup (which simulates the large-scale long-tailed scenario): (1) finetuning the previous TP's model outperforms training from scratch on data labeled with new ontology: 73.64% vs. 65.40% in accuracy; (2) jointly training on both old and new data significantly boosts accuracy to 82.98%; (3) taking advantage of the relationship between old and new labels (via Learning-with-Partial-Labels and SSL with pseudo-label refinement) further improves to 84.34%, effectively reaching an "upperbound" 84.51%!

**Contributions**. We make three major contributions. First, we motivate the problem LECO and define its benchmarking protocol. Second, we extensively study approaches to LECO by modifying methods from related problems, including semi-supervised learning, class-incremental learning, and learning with partial labels. Third, we draw consistent conclusions and technical insights, as described above.

## 2   Related Work

**Class-incremental learning** (CIL) [18, 47, 74, 85] and LECO both require learning classifiers for new classes over distinct time periods (TPs), but they have important differences. First, the typical CIL setup assumes new labels have no relations with the old [1, 47, 49], while LECO's new labels refine or expand the old / "coarse" labels (cf. Fig. 1). Second, CIL purposely sets a small memory buffer (for storing labeled data) and evaluates accuracy over both old and new classes with the emphasis on information retention or catastrophic forgetting [10, 40, 47, 82]. However, LECO allows all (history) labeled data to be stored and highlight the difficulty of learning new classes which are fine/subclasses of the old ones. Further, we note that many real-world applications store history data (and should not limit a buffer size to save data) for privacy-related consideration (e.g., medical data records) and as forensic evidence (e.g., videos from surveillance cameras). Therefore, to approach LECO, we apply CIL methods that do not restrict buffer size, which usually serve as upper bounds in CIL [17]. In this work, we repurpose "upper bound" CIL methods for LECO including finetuning [3, 80], joint training [11, 47, 50, 51], as well as the "lower bound" methods by freezing the backbone [3, 20, 66].

**Semi-supervised learning** (SSL) learns over both labeled and unlabeled data. State-of-the-art SSL methods follow a self-training paradigm [54, 65, 79] that uses a model trained over labeled data to pseudo-label the unlabeled samples, which are then used together with labeled data for training. For example, Lee *et al.* [44] use confident predictions as target labels for unlabeled data, and Rizve *et al.* [62] further incorporate low-probability predictions as negative pseudo labels. Some others improve SSL using self-supervised learning techniques [24, 83], which force predictions to be similar for different augmentations of the same data [6, 7, 68, 79, 84]. We leverage insights of SSL to approach LECO, e.g., pseudo labeling the old data. As pseudo labels are often biased [77], we further exploit the old-ontology to reject or improve the inconsistent pseudo labels, yielding better performance.

**Learning with partial labels** (LPL) tackles the case when some examples are fully labeled while others are only partially labeled [9, 15, 26, 53, 56, 87]. In the area of fine-grained recognition, partial labels can be coarse superclasses which annotate some training data [31, 45, 61, 73]. In a TP of LECO, the old data from previous TPs can be used as partially-labeled examples as they contain only coarse labels. Therefore, to approach LECO, we explore state-of-the-art methods [70] in this line of work. However, it is important to note that ontology evolution in LECO can be more than splitting old classes, e.g., the new class `special-subject` can be a result of merging multiple classes `child` and `police-officer`. Indeed, we find it happens quite often in the ontology evolution from the Mapillary's versioning from V1.2 [55] to V2.0 [52]. In this case, the LPL method does not show better results than the simple joint training approach (31.04 vs. 31.05 on Mapillary in Table 4).

## 3 Problem Setup of Learning with Evolving Class Ontology (LECO)

**Notations**. Among $T$ time periods (TPs), $TP^t$ has data distributions[3] $\mathcal{D}^t = \mathbf{X} \times \mathbf{Y}^t$, where $\mathbf{X}$ is the input and $\mathbf{Y}^t$ is the label set. The evolution of class ontology is ideally modeled as a tree structure: the $i^{th}$ class $\mathbf{y}_i^t \in \mathbf{Y}^t$ is a node that has a unique parent $\mathbf{y}_j^{t-1} \in \mathbf{Y}^{t-1}$, which can be split into multiple fine classes in $TP^t$. Therefore, $|\mathbf{Y}^t| > |\mathbf{Y}^{t-1}|$. $TP^t$ concerns classification w.r.t label set $\mathbf{Y}^t$.

**Benchmarks**. We define LECO benchmarks using three datasets: CIFAR100 [42], iNaturalist [69, 75], and Mapillary [52, 55] (Table 1). CIFAR100 is released under the MIT license, and iNaturalist and Mapillary are publicly available for non-commercial research and educational purposes. CIFAR100 and Mapillary contain classes related to person and potentially have fairness and privacy concerns, hence we cautiously proceed our research and release our code under the MIT License without re-distributing the data. The iNaturalist and Mapillary are large-scale and have long-tailed class distributions (refer to the Table **??** and **??** for class distributions). For each benchmark, we sample data from the corresponding dataset to construct time periods (TPs), e.g., $TP^0$ and $TP^1$ define their own ontologies of class labels $\mathbf{Y}^0$ and $\mathbf{Y}^1$, respectively. For *CIFAR-LECO*, we use the two-level hierarchy offered by the original CIFAR100 dataset [42]: using the 20 superclasses as the ontology in $TP^0$, each of them is split into 5 subclasses as new ontology in $TP^1$. For *iNat-LECO*, we used the most recent version of Semi-iNat-2021 [71] that comes with rich taxonomic labels at seven-level hierarchy. To construct two TPs, we choose the third level (i.e., "order") with 123 categories for $TP^0$, and the seventh ("species") with 810 categories for $TP^1$. We further construct four TPs with each having an ontology out of four at distinct taxa levels ["order", "family", "genus", "species"]. For Mapillary-LECO, we use its ontology of V1.2 [55] (2017) in $TP^0$, and that of V2.0 [52] (2021) in $TP^1$. It is worth noting that, as a real-world LECO example, Mapillary includes a catch-all `background` class (aka `void`) in V1.2, which was split into meaningful fine classes in V2.0, such as `temporary-barrier`, `traffic-island` and `void-ground`.

Moreover, in each TP of benchmark CIFAR-LECO and iNat-LECO, we randomly sample 20% data as validation set for hyperparameter tuning and model select. We use their official valsets as our test-sets for benchmarking. In Mapillary-LECO, we do not use a valset but instead use the default hyperparameters reported in [72], tuned to optimize another related dataset Cityscapes [14]. We find it unreasonably computational demanding to tune on large-scale semantic segmentation datasets. Table 1 summarizes our benchmarks' statistics.

**Remark: beyond class refining in ontology evolution**. In practice, ontology evolution also contains the case of class merging, as well as class renaming (which is trivial to address). The Mapillary's versioning clearly demonstrates this: 10 of 66 classes of its V1.2 were refined into new classes, resulting into 116 classes in total of its V2.0; objects of the same old class were either merged into a new class or assigned with new fine classes. Because of these non-trivial cases under the Mapillary-LECO benchmark, state-of-the-art methods of learning with partial labels are less effective but our other solutions work well.

**Remark: number of TPs and buffer size**. For most experiments, we set two TPs. One may think more TPs are required for experiments because CIL literature does so [18, 47, 74, 85]. Recall that CIL emphasizes catastrophic forgetting issue caused by using limited buffer to save history data, so using more TPs for more classes and limited buffer helps exacerbate this issue. Differently, LECO emphasizes the difficulty of learning new classes that are subclasses of the old ones, and does not

---

[3]The data distribution may not be necessarily static where LECO can still happen, as discussed in Section 1. Our work sets a static data distribution to simplify the study of LECO.

limit a buffer to store history data. Therefore, using two TPs are sufficient to show the difficulty. Even so, we have experiments that set four TPs, which demosntrate consistent conclusions drawn from those using two TPs. Moreover, even with unlimited buffer, CIL methods still work poorly on LECO benchmarks (cf. "TrainScratch" [59] and "FreezePrev" [47] in Table 2).

**Metric**. Our benchmarks study LECO through the lens of image classification (on CIFAR-LECO and iNat-LECO) and semantic segmentation (on Mapillary-LECO). Therefore, we evaluate methods using mean accuracy (mAcc) averaged over per-class accuracies, and mean intersection-over-union (mIoU) averaged over classes, respectively. The metrics treat all classes equally, avoiding the bias towards common classes due to the natural long-tailed class distribution (cf. iNaturalist [69, 75] and Mapillary [55]) (cf. class distributions in the appendix). We do not use specific algorithms to address the long-tailed recognition problem but instead adopt the recent simple technique to improve performance by tuning weight decay [2]. We run each method five times using random seeds on CIFAR-LECO and iNat-LECO, and report their averaged mAcc with standard deviations. For Mapillary, due to exoribitant training cost, we perform one run to benchmark methods.

## 4 Baseline Approaches to LECO

We first repurpose CIL and continual learning methods as baselines to approach LECO, as both require learning classifiers for new classes in TPs. We start with preliminary backgrounds.

**Preliminaries**. Following prior art [39, 47], we train convolutional neural networks (CNNs) and treat a network as a feature extractor $f$ (parametrized by $\theta_f$) plus a classifier $g$ (parameterized by $\theta_g$). The feature extractor $f$ consists of all the layers below the penultimate layer of ResNet [28], and the classifier $g$ is the linear classifier followed by softmax normalization. Specifically, in $TP^t$, an input image $x$ is fed into $f$ to obtain its feature representation $z = f(x; \theta_f)$. The feature $z$ is then fed into $g$ for the softmax probabilities, $q = g(z; \theta_g)$, w.r.t classes in $\mathbf{Y}^t$. We train CNNs by minimizing the Cross Entropy (CE) loss using mini-batch SGD. At $TP^t$, to construct a training batch $\mathcal{B}^t$, we randomly sample $K$ examples, i.e., $\mathcal{B}^t = \{(x_1, y_1^t), \cdots, (x_K, y_K^t)\}$. The CE loss on $\mathcal{B}^t$ is

$$\mathcal{L}(\mathcal{B}^t) = - \sum_{(x_k, y_k^t) \in \mathcal{B}^t} \mathcal{H}(y_k^t, g(f(x_k))) \tag{1}$$

where $\mathcal{H}(p, q) = - \sum_c p(c) log(q(c))$ is the entropy between two probability vectors.

Similary, for semantic segmentation, we use the same CE loss 1 at pixel level, along with the recent architecture HRNet with OCR module [72, 76, 81] (refer to appendix C for details).

**Annotation Strategies**. Closely related to LECO is the problem of class-incremental learning (CIL) or more generally, continual learning (CL) [47, 59, 60, 82, 85]. However, unlike typical CL benchmarks, LECO does not artificially limit the buffer size for storing history samples. Indeed, the expense of hard drive buffer is less of a problem compared to the high cost of data annotation. Therefore, LECO embraces all the history labeled data, regardless of being annoated using old or new ontologies. This leads to a fundamental question: whether to annotate new data or re-label the old, given a fixed labeling budget $N$ at each TP (cf. Fig. 1)?

- (**LabelNew**) Sample and annotate $N$ new examples using new ontology, which trivially infers the old-ontology labels. We further have the history data as labeled with old ontology.
- (**RelabelOld**) Re-annotate the $N$ history data using new ontology. However, these $N$ examples are all available data although they have both old- and new-ontology labels.

The answer is perhaps obvious – **LabelNew** – because it produces more labeled data. Furthermore, **LabelNew** allows one to exploit the old data to boost performance (Section 5). With proper techniques, this significantly reduces the performance gap with a model trained over both old and new data assumably annotated with new-ontology, termed as **AllFine** short for "supervised learning with *all fine* labels".

**Training Strategies**. We consider CL baseline methods as LECO baselines. When new tasks (i.e., classes of $\mathbf{Y}^t$) arrive in $TP^t$, it is desirable to transfer model weights trained for $\mathbf{Y}^{t-1}$ in $TP^{t-1}$ [47, 60]. To do so, we initialize weights $\theta_{f^t}$ of the feature extractor $f^t$ with $\theta_{f^{t-1}}$ trained in $TP^{t-1}$, i.e., $\theta_{f^t} := \theta_{f^{t-1}}$. Then we can

- (**FreezePrev**) Freeze $f^t$ to extract features, and learn a classifier $g^t$ over them [47].

Table 2: **Results of baseline methods for LECO** (more results in the Table **??**). For all our experiments, we run each method five times and report the averaged mean per-class accuracy/intersection-over-union (mAcc/mIoU in %) with standard deviations when available. For each benchmark, we bold the best mAcc/mIoU including ties that fall within its std. **FinetunePrev** is the best learning strategy that outperforms **TrainScratch** (cf. 73.64% vs. 65.69% with **LabelNew** on iNat-LECO). This implies that the coarse-to-fine evolved ontology serves as a good learning curriculum. **FreezePrev** underperforms **TrainScratch** (cf. 50.82% vs. 65.69% with LabelNew on iNat-LECO), confirming the difficulty of learning new classes that refine the old. **LabelNew** and **RelabelOld** achieve similar performance with the former using only new data but not the old altogether. If using both old and new data, we boost performance as shown in Table 3. The conclusions hold for varied number of training images and generalize to the Mapillary experiments, yet all methods fall short to the **AllFine** (which trains on all data assumed to be labeled with new ontology).

| Benchmark | #images / TP | TP$^0$ mAcc/mIoU | TP$^1$ Strategy | TP$^1$ mAcc/mIoU | | |
|---|---|---|---|---|---|---|
| | | | | LabelNew | RelabelOld | AllFine |
| *CIFAR-LECO* | 10000 | $77.78 \pm 0.27$ | FinetunePrev | $\mathbf{65.83 \pm 0.53}$ | $\mathbf{65.30 \pm 0.32}$ | $71.30 \pm 0.41$ |
| | | | FreezePrev | $52.64 \pm 0.50$ | $52.60 \pm 0.52$ | $53.79 \pm 0.43$ |
| | | | TrainScratch | $\mathbf{65.40 \pm 0.53}$ | $64.98 \pm 0.34$ | $71.43 \pm 0.24$ |
| *iNat-LECO* | 50000 | $64.21 \pm 1.61$ | FinetunePrev | $\mathbf{73.64 \pm 0.46}$ | $71.26 \pm 0.54$ | $84.51 \pm 0.66$ |
| | | | FreezePrev | $50.82 \pm 0.68$ | $54.60 \pm 0.49$ | $54.08 \pm 0.31$ |
| | | | TrainScratch | $65.69 \pm 0.46$ | $65.78 \pm 0.56$ | $73.10 \pm 0.80$ |
| *Mapillary-LECO* | 2500 | 37.01 | FinetunePrev | **30.39** | 29.05 | 31.73 |
| | | | TrainScratch | 27.24 | 27.21 | 27.73 |

- (**FinetunePrev**) Finetune $f^t$ and learn a classifier $g^t$ altogether [47].
- (**TrainScratch**) Train $f^t$ from random weights [59].

**Results**. We experiment with different combinations of data annotation and training strategies in Table 2. We adopt standard training techniques including SGD with momentum, cosine annealing learning rate schedule, weight decay and RandAugment [16]. We ensure the maximum training epochs (2000/300/800 on CIFAR/iNat/Mapillary respectively) to be large enough for convergence. See Table 2-caption for salient conclusions.

## 5 Improving LECO with the Old-Ontology Data

The previous section shows that **FinetunePrev** is the best training strategy, and **LabelNew** and **RelabelOld** achieve similar performance with the former being not exploiting the old data. In this section, we stick to **FinetunePrev** and **LabelNew** and study how to exploit the old data to improve LECO. To do so, we repurpose methods from semi-supervised learning and incremental learning.

**Notation:** We refer to the new-ontology (fine) labeled samples at TP$^t$ as $S^t \subset \mathbf{X} \times \mathbf{Y}^t$, and old-ontology (coarse) labeled samples accumulated from previous TPs as $S^{1:t-1} = S^1 + \cdots + S^{t-1}$.

### 5.1 Exploiting the Old-Ontology Data via Semi-Supervised Learning (SSL)

State-of-the-art SSL methods aim to increase the amount of labeled data by pseudo-labeling the unlabeled examples [25, 70]. Therefore, a natural thought to utilize the old examples is to *ignore their old-ontology labels* and use SSL methods. Below, we describe several representative SSL methods which are developed in the context of image classification.

Each input sample $x$, if not stated otherwise, undergoes a strong augmentation (RandAugment [16] for classification and HRNet [72] augmentation for segmentation) denoted as $\mathcal{A}(x)$ before fed into $f$. For all SSL algorithms, we construct a batch $\mathcal{B}_K$ of $K$ samples from $S_t$ and a batch $\hat{\mathcal{B}}_M$ of $M$ samples from $S^{1:t-1}$. For brevity, we hereafter ignore the superscript (TP index). On new-ontology labeled samples, we use Eq. (1) to obtain $\mathcal{L}(\mathcal{B}_K)$, and sum it with an extra SSL loss on $\hat{\mathcal{B}}_M$, denoted as $\mathcal{L}_{SSL}(\hat{\mathcal{B}}_M)$. We have four SSL losses below.

Following [70], we use "Self-Training" to refer to a teacher-student distillation procedure [29]. Note that we adopt the **FinetunePrev** strategy and so both teacher and student models are initialized from the previous TP's weights. We first have two types of Self-Training:

- **ST-Hard** (Self-Training with one-hot labels): We first train a teacher model ($f_{teacher}$ and $g_{teacher}$) on the training set $S^t$, then use the teacher to "pseudo-label" history data in $S^{1:t-1}$. Essentially, for a $(x_m, y_m) \in S^{1:t-1}$, the old-ontology label $y_m$ is ignored and a one-hot hard label is produced as $q_h = \arg\max g_{teacher}(f_{teacher}(\mathcal{A}(x_m)))$. Finally, we train a student model ($f_{student}$ and $g_{student}$) with CE loss:

$$\mathcal{L}_{SSL}(\hat{\mathcal{B}}_M) = \sum_{(x_m, \cdot) \in \hat{\mathcal{B}}_M} \mathcal{H}(q_h, g_{student}(f_{student}(\mathcal{A}(x_m)))) \qquad (2)$$

- **ST-Soft** (Self-Training with soft labels): The teacher is trained in same way as **ST-Hard**, but the difference is that pseudo-labels are softmax probability vectors $q_s = g_{teacher}(f_{teacher}(\mathcal{A}(x_m)))$ instead. The final loss is the KL divergence [58, 70]:

$$\mathcal{L}_{SSL}(\hat{\mathcal{B}}_M) = \sum_{(x_m, \cdot) \in \hat{\mathcal{B}}_M} \mathcal{H}(q_s, g_{student}(f_{student}(\mathcal{A}(x_m)))) \qquad (3)$$

In contrast to the above SSL methods that train separate teacher and student models, the following two train a single model along with the pseudo-labeling technique.

- **PL (Pseudo-Labeling)** [44] pseudo-labels image $x_m$ as $q = g(f(\mathcal{A}(x_m)))$ using the current model. It also filters out pseudo-labels that have probabilities less than 0.95.

$$\mathcal{L}_{SSL}(\hat{\mathcal{B}}_M) = \sum_{(x_m, \cdot) \in \hat{\mathcal{B}}_M} \mathbb{1}[\max q > 0.95]\mathcal{H}(q, g(f(\mathcal{A}(x_m))) \qquad (4)$$

- **FixMatch** [68] imposes a consistency regularization into PL by enforcing the model to produce the same output across weak augmentation ($a(x_m)$, usually standard random flipping and cropping) and strong augmentation ($\mathcal{A}(x_m)$, in our case is weak augmentation followed by RandAugment [16]). The pseudo-label is produced by weakly augmented version $q_w = g(f(a(x_m)))$. The final loss is:

$$\mathcal{L}_{SSL}(\hat{\mathcal{B}}_M) = \sum_{(x_m, \cdot) \in \hat{\mathcal{B}}_M} \mathbb{1}[\max q_w > 0.95]\mathcal{H}(q_w, g(f(\mathcal{A}(x_m))) \qquad (5)$$

While the above techniques are developed in the context of image classification, they appear to be less effective for semantic segmentation (on the Mapillary dataset), e.g., ST-Soft requires extremely large storage to save per-pixel soft labels, FixMatch requires two large models designed for semantic segmentation. Therefore, in Mapillary-LECO, we adopt the **ST-Hard** which is computationally friendly given our compute resource (Nvidia GTX-3090 Ti with 24GB RAM).

## 5.2  Exploiting the Old-Ontology Data via Joint Training

The above SSL approaches discard the old-ontology labels, which could be a coarse supervision in training. A simple approach to leverage this coarse supervision is **Joint Training**, a classic incremental learning method [18, 47] that trains jointly on both new and old tasks. Interestingly, **Joint Training** is usually the upper bound performance in traditional incremental learning benchmarks [47]. Therefore, we perform **Joint Training** to learn a shared feature extractor $f$ and independent classifiers $(g^1, \cdots, g^t)$ w.r.t both new and old ontologies. We define the following loss at $TP^t$ using old-ontology data from $TP^1$ to $TP^{t-1}$:

$$\mathcal{L}_{old}(\hat{\mathcal{B}}_M) = \sum_{t' \in \{1, \cdots, t-1\}} \sum_{(x_m, y_m^{t'}) \in \hat{\mathcal{B}}_M^{t'}} \mathcal{H}(y_m^{t'}, g^{t'}(f(\mathcal{A}(x_m))) \qquad (6)$$

where $y_m^{t'}$ is the coarse label of $x_m$ at $TP^{t'}$ and $\hat{\mathcal{B}}_M^{t'}$ is the subset of $\hat{\mathcal{B}}_M$ with label $y_m^{t'}$ available. Importantly, because new-ontology fine labels $\mathbf{y}_i^t \in \mathbf{Y}^t$ refine old-ontology coarse labels, e.g., $\mathbf{y}_j^{t-1} \in \mathbf{Y}^{t-1}$, we can trivially infer *the coarse label for any fine label*. Therefore, we also apply $\mathcal{L}_{old}$ for fine-grained samples in $B_K$, i.e., $\mathcal{L}_{old}(\mathcal{B}_K)$. We have the final loss for **Joint Training** as $\mathcal{L}_{Joint} = \mathcal{L}_{old}(\mathcal{B}_K) + \mathcal{L}_{old}(\hat{\mathcal{B}}_M)$.

## 5.3  Results

Table 3 compares the above-mentioned SSL and Joint Training methods. In training, we sample the same amount of old data and new data in a batch, i.e., $|\hat{\mathcal{B}}_M| = |\mathcal{B}_K|$ (64 / 30 / 4 for CIFAR / iNat / Mapillary). We assign equal weight to $\mathcal{L}_{SSL}$, $\mathcal{L}_{Joint}$, and $\mathcal{L}$. See Table 3-caption for conclusions.

Table 3: **Results of SSL and Joint Training methods**. Following the notation of Table 2, we study **LabelNew** annotation strategy that make use of algorithms for learning from old examples *without* exploiting the relationship between old-vs-new labels. Recall that AllFine uses $\mathcal{L}$ only to train on all the data labeled with new ontology (doubling annotation cost). Both $\mathcal{L}_{Joint}$ and the best $\mathcal{L}_{SSL}$ (ST-Soft) outperform the baseline which uses loss $\mathcal{L}$ only (i.e., LabelNew + FinetunePrev in Table 2). ST-Soft yields the most performance gain. Using all the three losses together consistently improves mAcc / mIoU for all the benchmarks, confirming the benefit of exploiting the old examples, e.g., on *iNat-LECO*, this boosts the performance to 83.6%, approaching the AllFine (84.5%)!

| Benchmark | #images / TP | SSL Alg | TP¹ mAcc/mIoU | | | | |
| --- | --- | --- | --- | --- | --- | --- | --- |
| | | | $\mathcal{L}$ only | $+\mathcal{L}_{SSL}$ | $+\mathcal{L}_{Joint}$ | $+\mathcal{L}_{Joint}$ $+\mathcal{L}_{SSL}$ | AllFine |
| *CIFAR-LECO* | 10000 | ST-Hard | $65.83 \pm 0.27$ | $67.11 \pm 0.34$ | $68.74 \pm 0.21$ | $67.97 \pm 0.37$ | $71.30 \pm 0.41$ |
| | | ST-Soft | | $\mathbf{69.86 \pm 0.30}$ | | $\mathbf{69.77 \pm 0.71}$ | |
| | | PL | | $65.86 \pm 0.14$ | | $68.43 \pm 0.40$ | |
| | | FixMatch | | $67.13 \pm 0.36$ | | $69.03 \pm 0.43$ | |
| *iNat-LECO* | 50000 | ST-Hard | $73.64 \pm 0.46$ | $75.23 \pm 0.14$ | $82.98 \pm 0.33$ | $78.23 \pm 0.38$ | $84.51 \pm 0.66$ |
| | | ST-Soft | | $79.64 \pm 0.41$ | | $\mathbf{83.61 \pm 0.39}$ | |
| | | PL | | $73.56 \pm 0.09$ | | $82.87 \pm 0.51$ | |
| | | FixMatch | | $74.57 \pm 0.54$ | | $83.00 \pm 0.33$ | |
| *Mapillary-LECO* | 2500 | ST-Hard | 30.39 | 30.06 | **31.05** | **31.09** | 31.73 |

# 6 Improving LECO with Taxonomic Hierarchy

While the SSL approaches exploit old samples and Joint Training further utilizes their old-ontology labels, none of them fully utilize the given taxonomic relationships between old and new ontologies. The fact that each fine-grained label $\mathbf{y}_i^t \in \mathbf{Y}^t$ refines a coarse label $\mathbf{y}_j^{t-1} \in \mathbf{Y}^{t-1}$ suggests a chance to improve LECO by exploiting such taxonomic relationship between the old and new classes [58, 69].

## 6.1 Incorporating Taxonomic Hierarchy via Learning-with-Partial-Labels (LPL)

**Joint Training** trains separate classifiers for each TP. This is suboptimal because we can simply marginalize fine-classes' probabilities for coarse-classes' probabilities [19]. This technique has been used in other works [32, 69] and is commonly called Learning-with-Partial-Labels (LPL) [32]. Concretely, the relationship between classes in $\mathbf{Y}^t$ and $\mathbf{Y}^{t'}$ with $t > t'$ can be summarized by an edge matrix $\mathbf{E}_{t \to t'}$ with shape $|\mathbf{Y}^t| \times |\mathbf{Y}^{t'}|$. $\mathbf{E}_{t \to t'}[i, j] = 1$ when an edge $y_i^t \to y_j^{t'}$ exists in the taxonomy tree, i.e., $y_j^{t'}$ is the superclass of $y_i^t$; otherwise $\mathbf{E}_{t \to t'}[i, j] = 0$. Therefore, to obtain prediction probability of a coarse-grained label $y^{t'}$ for a sample $x$, we sum up the probabilities in $g^t(f^t(x))$ that correspond to fine labels of $y^{t'}$, which can be efficiently done by a matrix vector multiplication: $g^t(f^t(x)) \cdot \mathbf{E}_{t \to t'}$. Formally, we define a loss over a sampled batch $\hat{\mathcal{B}}_M \subset S^{1:t-1}$ as:

$$\mathcal{L}_{OldLPL}(\hat{\mathcal{B}}_M) = \sum_{t' \in \{1, \cdots, t-1\}} \sum_{(x_m, y_m^{t'}) \in \hat{\mathcal{B}}_M^{t'}} \mathcal{H}(y_m^{t'}, g^t(f^t(\mathcal{A}(x_m))) \cdot \mathbf{E}_{t \to t'}) \tag{7}$$

We apply this loss on the fine-labeled data and define the **LPL Loss** $\mathcal{L}_{LPL} = \mathcal{L}_{oldLPL}(\mathcal{B}_K) + \mathcal{L}_{oldLPL}(\hat{\mathcal{B}}_M)$.

## 6.2 Incorporating Taxonomic Hierarchy via Pseudo-label Refinement

Another strategy to incorporate the taxonomic hierarchy is to exploit coarse labels in the pseudo-labeling process, which has been explored in a different task (semi-supervised few-shot learning) [58]. Prior art shows that the accuracy of pseudo labels is positively correlated with SSL performance [68]. Therefore, at $\text{TP}^t$, in pseudo-labeling an old sample $x_m$, we use its coarse label $y_m^{t'}$ ($t' < t$) to clean up inconsistent pseudo labels. In particular, we consider

1. **Filtering**: we filter out (reject) pseudo-labels that do not align with the ground-truth coarse labels, i.e., rejecting pseudo-label $q = g^t(f^t(x))$ if $\mathbf{E}_{t \to t'}[\arg \max_c q_c, y_m^{t'}] = 0$.
2. **Conditioning**: we clean up the pseudo-label $q$ by zeroing out all scores unaligned with the true coarse label: $q[c] := q[c] * \mathbf{E}_{t \to t'}[c, y_m^{t'}]$. We then renormalize $q[c] := q[c] / \sum_i q[i]$.

Table 4: **Results of exploiting taxonomic hierarchy** (accuracy in %). We copy the best results in Table 3 (all obtained by $+\mathcal{L}_{Joint} +\mathcal{L}_{SSL}$) for reference. Adding supervision via taxonomic hierarchy is effective: (1) simply using $\mathcal{L}_{LPL}$ already matches the previously best performance (by by $+\mathcal{L}_{Joint} +\mathcal{L}_{SSL}$); (2) adding SSL loss along with pseudo-label refinement via either **Filtering** or **Conditioning** further bridges the gap to AllFine. As a result, for example, on *iNat-LECO*, the best strategy ($+\mathcal{L}_{LPL} + \mathcal{L}_{SSL/Cond}$ with ST-Hard) is only 0.2% lower than the AllFine!

| Benchmark | #images / TP | SSL Alg | TP[1] mAcc | | | | |
| --- | --- | --- | --- | --- | --- | --- | --- |
| | | | $+\mathcal{L}_{Joint}$ $+\mathcal{L}_{SSL}$ | $+\mathcal{L}_{LPL}$ | $+\mathcal{L}_{LPL}$ $+\mathcal{L}_{SSL/Filter}$ | $+\mathcal{L}_{LPL}$ $+\mathcal{L}_{SSL/Cond}$ | AllFine |
| *CIFAR-LECO* | 10000 | ST-Hard | $69.77 \pm 0.71$ | $69.31 \pm 0.30$ | $69.90 \pm 0.29$ | $\mathbf{70.42 \pm 0.23}$ | $71.30 \pm 0.41$ |
| | | ST-Soft | | | $70.10 \pm 0.12$ | $70.02 \pm 0.63$ | |
| | | PL | | | $69.20 \pm 0.20$ | $69.60 \pm 0.26$ | |
| | | FixMatch | | | $69.61 \pm 0.17$ | $69.85 \pm 0.46$ | |
| *iNat-LECO* | 50000 | ST-Hard | $83.61 \pm 0.39$ | $83.62 \pm 0.21$ | $\mathbf{84.23 \pm 0.61}$ | $\mathbf{84.34 \pm 0.25}$ | $84.51 \pm 0.66$ |
| | | ST-Soft | | | $\mathbf{84.12 \pm 0.31}$ | $83.76 \pm 0.40$ | |
| | | PL | | | $83.72 \pm 0.22$ | $\mathbf{84.16 \pm 0.19}$ | |
| | | FixMatch | | | $83.91 \pm 0.39$ | $83.95 \pm 0.40$ | |
| *Mapillary-LECO* | 2500 | ST-Hard | $31.09$ | $31.04$ | $\mathbf{31.41}$ | $31.23$ | $31.73$ |

Table 5: **Baseline results on iNat-4TP-LECO** (accuracy in %). **FinetunePrev** significantly and consistently outperforms **TrainScratch**, suggesting representation trained for coarse-grained classification can serve as powerful initialization for finetning in later TPs. Furthermore, **LabelNew** and **RelabelOld** achieve almost the same performance across all TPs, but we will show in Table 6 that once exploiting the old-ontology data and labels, we can significantly improve the performance, approaching the "upperbound" **AllFine**.

| Benchmark | #images / TP | TP[0] mAcc | Training Strategy | Annotation | TP[1] mAcc | TP[2] mAcc | TP[3] mAcc |
| --- | --- | --- | --- | --- | --- | --- | --- |
| *iNat-4TP-LECO* | 25000 | $46.6 \pm 1.2$ | TrainScratch | LabelNew | $50.9 \pm 0.6$ | $49.4 \pm 0.6$ | $48.4 \pm 0.6$ |
| | | | | RelabelOld | $50.1 \pm 1.1$ | $49.7 \pm 0.4$ | $47.9 \pm 0.6$ |
| | | | | AllFine | $62.7 \pm 0.5$ | $63.8 \pm 0.3$ | $58.0 \pm 0.2$ |
| | | | FinetuneNew | LabelNew | $55.1 \pm 0.7$ | $\mathbf{61.7 \pm 0.8}$ | $\mathbf{63.2 \pm 0.4}$ |
| | | | | RelabelOld | $\mathbf{55.6 \pm 0.9}$ | $\mathbf{61.7 \pm 0.9}$ | $63.1 \pm 0.4$ |
| | | | | AllFine | $67.7 \pm 0.6$ | $78.8 \pm 0.4$ | $84.6 \pm 0.3$ |

## 6.3 Results

Table 4 shows the results of **LPL Loss** and SSL strategies with pseudo-label refinement. We can see that simply adding the **LPL Loss** $\mathcal{L}_{LPL}$ matches the SOTA (achieved by naive SSL and **Joint Training** strategies in Table 3). Also, adding SSL loss along with pseudo-label refinement via either **Filtering** or **Conditioning** further bridges the gap to **AllFine**. For example, on *iNat-LECO*, the best strategy ($+\mathcal{L}_{LPL} + \mathcal{L}_{SSL/Cond}$ with ST-Hard) is only 0.2% lower than the **AllFine** (84.3% v.s. 84.5%). We acknowledge that the **AllFine** can be improved further if using LPL loss as well [8] but we use the current version just as a reference of performance. We also tried combining $\mathcal{L}_{SSL/Filter}$ or $\mathcal{L}_{SSL/Cond}$ with the **Joint Training**, and observe similar improvements upon $\mathcal{L}_{SSL} + \mathcal{L}_{Joint}$ (results in Table **??** and **??**).

# 7 Further Experiments with More Time Periods

We further validate our proposed approahces with more TPs. In this experiment, we construct a benchmark *iNat-4TP-LECO* using iNaturalist by setting four TPs, each having an ontology at distinct taxa levels: "order" (123 classes), "family" (339 classes), "genus" (729 classes), "species" (810 classes). We show in Table 5 that **FinetuneNew** outperforms **TrainScratch** from TP[1] to TP[3]. Furthermore, we study **LabelNew** annotation strategy with **FinetunePrev** training strategy (as in Table 3 and 4) for *iNat-4TP-LECO*, and report results in Table 6. In summary, all the proposed solutions generalize to more than two TPs, and simple techniques that utilize old-ontology labels (such as $\mathcal{L}_{Joint}$) and label hierarchy (such as $\mathcal{L}_{LPL}$) achieve near-optimal performance compared to **AllFine** "upperbound".

Table 6: **Results of iNat-4TP-LECO with FinetuneNew training strategy and LabelNew annotation strategy** (accuracy in %). We select two best performing SSL methods on **iNat-LECO** in Table 3, **FixMatch** and **ST-Soft**, for benchmarking. Clearly, all solutions proposed in the paper achieve consistent gains over baseline methods (Table 5). First, leveraging old-ontology data via SSL (e.g., $+\mathcal{L}_{\text{ST-Soft}}$) can improve the classification accuracy from 63.2% to 67.6% at TP$^3$. Also, utilizing the old-ontology labels via simple joint training can boost performance to 83.0%. Lastly, incorporating taxonomic hierarchy through LPL and SSL (e.g., $+\mathcal{L}_{LPL} + \mathcal{L}_{\text{ST-Soft/Filter}}$) further improves to 85.3%.

| Benchmark | #images / TP | Combination of losses | TP$^1$ mAcc | TP$^2$ mAcc | TP$^3$ mAcc |
|---|---|---|---|---|---|
| | | AllFine | $67.7 \pm 0.6$ | $78.8 \pm 0.4$ | $84.6 \pm 0.3$ |
| | | $\mathcal{L}$ only | $55.1 \pm 0.7$ | $61.7 \pm 0.8$ | $63.2 \pm 0.4$ |
| | | $+\mathcal{L}_{\text{FixMatch}}$ | $56.7 \pm 1.0$ | $61.4 \pm 1.0$ | $65.0 \pm 0.9$ |
| | | $+\mathcal{L}_{\text{ST-Soft}}$ | $59.6 \pm 1.2$ | $67.8 \pm 1.2$ | $67.6 \pm 1.3$ |
| *iNat-4TP-LECO* | 25000 | $+\mathcal{L}_{Joint}$ | $64.7 \pm 0.6$ | $74.3 \pm 0.9$ | $82.4 \pm 0.9$ |
| | | $+\mathcal{L}_{Joint} + \mathcal{L}_{\text{FixMatch}}$ | $64.6 \pm 0.8$ | $75.5 \pm 0.7$ | $82.8 \pm 0.7$ |
| | | $+\mathcal{L}_{Joint} + \mathcal{L}_{\text{ST-Soft}}$ | $64.4 \pm 1.4$ | $76.1 \pm 1.5$ | $83.0 \pm 1.5$ |
| | | $+\mathcal{L}_{LPL}$ | $67.5 \pm 0.9$ | $76.7 \pm 0.9$ | $83.6 \pm 1.0$ |
| | | $+\mathcal{L}_{LPL} + \mathcal{L}_{\text{FixMatch/Filter}}$ | $\mathbf{67.7 \pm 0.8}$ | $77.0 \pm 0.9$ | $\mathbf{85.3 \pm 0.6}$ |
| | | $+\mathcal{L}_{LPL} + \mathcal{L}_{\text{FixMatch/Cond}}$ | $67.5 \pm 0.7$ | $71.0 \pm 0.8$ | $78.8 \pm 0.6$ |
| | | $+\mathcal{L}_{LPL} + \mathcal{L}_{\text{ST-Soft/Filter}}$ | $\mathbf{67.7 \pm 1.1}$ | $\mathbf{77.3 \pm 1.1}$ | $85.1 \pm 1.2$ |
| | | $+\mathcal{L}_{LPL} + \mathcal{L}_{\text{ST-Soft/Cond}}$ | $66.3 \pm 1.0$ | $72.7 \pm 1.3$ | $79.4 \pm 1.2$ |

## 8 Discussions and Conclusions

We introduce the problem *Learning with Evolving Class Ontology* (LECO) motivated by the fact that lifelong learners must recognize concept vocabularies that evolve over time. To explore LECO, we set up its benchmarking protocol and study various approaches by repurposing methods of related problems. Extensive experiments lead to several surprising conclusions. First, we find the status-quo for dataset versioning, where old data is relabeled, is not the most effective strategy. One should label new data with new labels, but the mixture of old and new labels is effective only by exploiting semi-supervised methods that exploit the relationship between old and new labels. Surprisingly, such strategies can approach an upperbound of learning from an oracle aggregate dataset assumably annotated with new labels.

*Societal Impacts*. Studying LECO has various positive impacts as it reflects open-world development of machine-learning solutions. For example, LECO emphasizes lifelong learning and maintainance of machine-learning systems, which arise in inter-disciplinary research (where a bio-image analysis system needs to learn from a refined taxonomy, cf. iNaturalist), and autonomous vehicles (where critical fine-grained classes appear as operations expand to more cities). We are not aware of negative societal impacts of LECO over those that arise in traditional image classification.

*Future work to address limitations*. We notice several limitations and anticipate future work to address them. Firstly, the assumption of constant annotation cost between the old and new data needs to be examined, but this likely requires the design of novel interfaces for (re)annotation. Secondly, our work did not consider the cost for obtaining new data, which might include data mining and cleaning. Yet, in many applications, data is already being continuously collected (e.g., by autonomous fleets and surveillance cameras). This also suggests that LECO incorporate unlabeled data. Moreover, perhaps surprisingly, finetuning previous TP's model significantly outperforms training from scratch. This suggests that the coarse-to-fine ontology evolution serve as a good curriculum of training, a topic widely studied in cognition, behavior and psychology [43, 57, 67]. Future work should investigate this, and favorably study efficiently finetuning algorithms [13, 30, 38, 46, 86]. Lastly, we explore LECO on well-known benchmarks with static data distributions over time periods; in the future, we would like to embrace temporal shifts in the data distribution as well [49].

## Acknowledgments and Disclosure of Funding

This research was supported by CMU Argo AI Center for Autonomous Vehicle Research.

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
