**Appendix**



## A: In LECO, label new data or relabel the old?

In LECO, the foremost question to answer is whether to label new data or relabel the old (Fig. 2-right). As discussed in the main paper, both labeling protocols have been used in the community. For example, in the context of autonomous driving research, Argoverse labeled new data with new ontology in its updated version (from V1.0 [12] to V2.0 [78]), whereas Mapillary related the old data from its V1.2 [55] to V2.0 [52]. Our extensive experiments convincingly demonstrate that a better strategy is to label new data, simply because doing so provides more (labeled) data.

Data acquisition (for new data) might be costly. However, many real-world applications acquire data continuously regardless of the cost. For example, autonomous vehicle fleets collect data continuously, so do surveillance cameras that record video frames. Therefore, labeling such new data is reasonable in the real world.

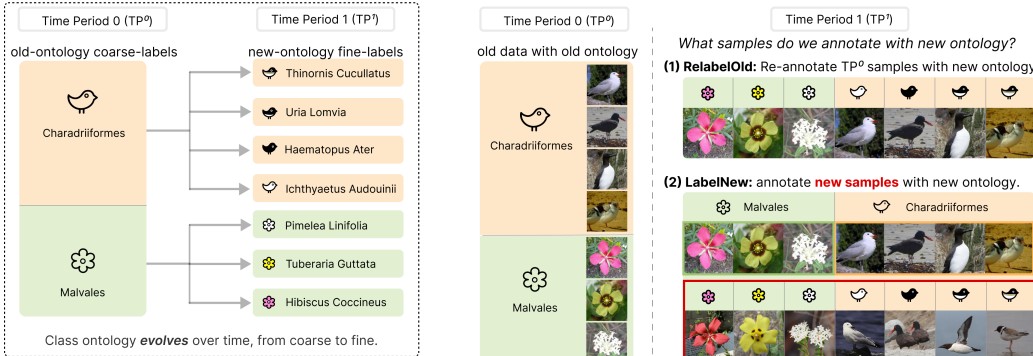

Figure 2: **Left:** Learning with Evolving Class Ontology (LECO) requires training models in time periods (TPs) that refine old ontologies in a coarse-to-fine fashion. This leads to a basic question: for the next TP, should one relabel the old data or label new data? Interestingly, both labeling protocols have been used in the community for large-scale datasets. **Right:** Our extensive experiments provide the (perhaps obvious) answer – one should always annotate *new* data with the *new* ontology rather than re-annotating the *old*. One reason is that the former produces more labeled data. Following this labeling protocol and to address LECO, we leverage insights from semi-supervised learning and learning-with-partial-labels to learn from such heterogenous annotations, approaching the upper bound of learning from an oracle aggregate dataset with all new labels.

## B: Rationale behind the selection of levels on iNaturalist

We repurpose the iNaturalist dataset to set up a LECO benchmarking protocol. The dataset has seven levels of taxa, allowing to use such as superclasses in LECO's time periods. To determine the levels to use, we have two principles aiming to have a clean and challenging setup to study LECO. First, we think it is good to have more fine-grained classes in TP1, hence we choose the most fine-grained level "species" in TP1. Second, we think all coarse-classes should be split later to emphasize the difficulty of learning with class-evolution, and TP0's classification task should be challenging as well with more classes. Therefore, we choose the "order" level which has 123 taxa. As reference, the original iNat dataset has seven levels: kingdom (3 taxa), phylum (8), class (29), order (123), family (339), genus (729), and species (810). The long-tailed class distributions in the two TPs are depicted in Fig. 4.

## C: Experiments on Mapillary V1.2 → V2.0

As we briefly discussed in the main paper, Mapillary is a real-world example where a large dataset evolved over time from V1.2 [55] to V2.0 [52]. We include additional experiments on this real-world dataset versioning scenario. Mapillary is a rich and diverse street-level imagery dataset that was

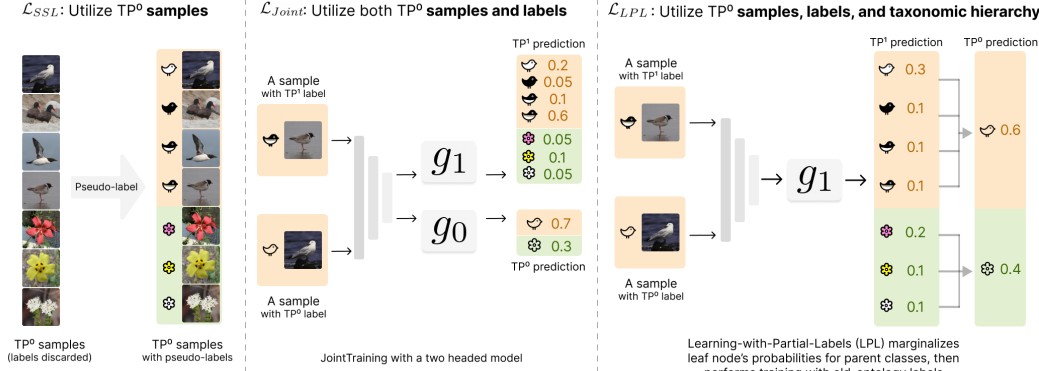

Figure 3: **Visual illustration of LECO strategies**. **Left:** $\mathcal{L}_{SSL}$ generates pseudo new-ontology/fine labels for old-ontology samples via semi-supervised learning (SSL). However, this ignores old-ontology/coarse labels which might serve as coarse-level supervision. We therefore advocate for $\mathcal{L}_{SSL/Filter}$ that filters out pseudo-labels with wrong coarse labels, or $\mathcal{L}_{SSL/Cond}$ that conditions leave's probabilities based on the ground-truth parent/coarse class. **Middle:** $\mathcal{L}_{Joint}$ utilizes the old-ontology labels that come with the samples; to reconcile for different labels at different TPs, it trains a model with multiple classification heads. **Right:** $\mathcal{L}_{LPL}$ (learning-with-partial-labels) further utilizes the taxonomic hierarchy by exploiting the fact that newly added classes in LECO are refined from old classes. It does so by marginalizing leaves' probabilties for their parent classes.

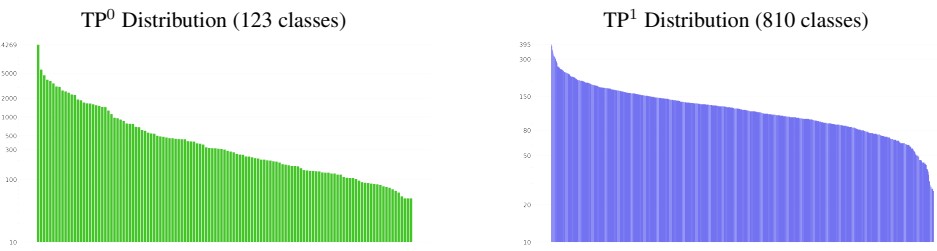

*iNat-LECO* Per-Class Image Distributions (Y-axis is log-scaled)

Figure 4: **iNat-LECO Per-Class Distributions.** We plot distributions of training images for *iNat-LECO* in log-scale. The x-axis are classes sorted by the number of images. The y-axis shows the number of images per class.

collected for semantic segmentation research in the context of autonomous driving. We repurpose this dataset for setting a LECO benchmark. Briefly, we use the ontology of Mapillary V1.2 in time period 1 (TP1), and V2.0 in TP2. TP1 has 66 classes including a catch-all `background` (aka `void`) class. TP1 (i.e., Mapillary V2.0 [4]) contains 124 classes, out of which 116 are used for evaluation.

**Mapillary-LECO benchmark**. The original Mapillary contains 18k training images, which is much larger than other mainstream street-level segmentation datasets such as Cityscape [14] (which has 2975 training and 500 validation images). In this work, we repurpose Mapillary to set up a LECO benchmark by splitting training set into two subsets (each containing 2.5k or 9k images) for the two time periods. We use the original validation set of Mapillary with 2k images as our test set. The detailed statistics is shown in Table 7.

**Baseline Experiments on Mapillary-LECO**. We used the state-of-the-art architecture HRNet-V2-W48 for semantic segmentation with OCR module [72, 76, 81]. We noticed that for semantic segmentation on street-level imagery datasets [14], it is common to start from a Imagenet-pretrained model. Hence we name the weight initialization schemes for *Mapillary-LECO* as:

- (**TrainRandom**) Train an entire network from randomly initialized weights.
- (**TrainScratch**) Initialize the model backbone with ImageNet-pretrained weights.[5]

As we can see from Table 8, **TrainScratch** (from ImageNet pretrained weights) as a popular initialization strategy for segmentation tasks [14, 55] indeed performs better than **TrainRandom**. Therefore,

---

[4] https://blog.mapillary.com/update/2021/01/18/vistas-2-dataset.html
[5] Weights are available at https://github.com/HRNet/HRNet-Semantic-Segmentation

Table 7: **Mapillary-LECO statistics**. We repurpose Mapillary [52, 55] V1.2 to V2.0 (a real-world dataset versioning scenario) to set up a LECO benchmark. Since V2.0 contains the same set of images as V1.2, in order to simulate a scenario with new samples, we select the first 5k or all 18k training images, then split them to 2.5k/9k for $TP^0$ and 2.5k/9k for $TP^1$. Note that in $TP^0$, the 66 classes include a catch-all `background` (or `void`) class, which is subsequently refined to 9 fine-grained classes such as `traffic-cone` and `traffic-island`.

| dataset | Time Period 0 ($TP^0$) | | | Time Period 1 ($TP^1$) | | |
|---|---|---|---|---|---|---|
| | #classes | #train | #test | #classes | #train | #test |
| *Mapillary-LECO* | 66 | 2.5k/9k | 2k | 116 | 2.5k/9k | 2k |

for our **FinetunePrev** strategy, we finetune the model checkpoint obtained via **TrainScratch** strategy on $TP^0$ data. To further bridge the gap to **AllFine**, we experiment other advanced techniques introduced in this paper such as $\mathcal{L}_{SSL}$ and $\mathcal{L}_{Joint}$ as shown in Table 10.

**Results of various LECO strategies**. Since Mapillary does not reveal the taxonomic hierarchy from V1.2 to V2.0, we use the ground-truth label maps to automatically retrace the ontology evolution. Because Mapillary adopts **RelabelOld** strategy, each input image has both V1.2 and V2.0 label map. We exploit this fact to determine a single parent class in V1.2 for each of the 116 classes in V2.0 following a simple procedure (taking `bird` class in V2.0 as an example):

- First, we use the ground-truth V2.0 label maps to find all pixels labeled as `bird`.
- Then, we locate those pixels in V1.2 label maps, and choose the V1.2 class that occupies the most pixels as `bird`'s parent.

We find that this simple procedure produces a reliable taxonomic hierarchy that can be used to improve the performance via $\mathcal{L}_{LPL}$, $\mathcal{L}_{SSL/Filter}$ or $\mathcal{L}_{SSL/Cond}$, as shown in Table 11. Note that this procedure does not model class merging in Mapillary; however, our solutions still remain effective.

**Segmentation Loss**. We make use of a standard pixel-level cross-entropy loss function for training on Mapillary. One difference from our previous image classification setup is that both the ontology and semantic label maps changed from V1.2 to V2.0. In general, we find the label maps on V2.0 to be of higher quality. As such, we (1) do not add coarse supervision on $TP^1$ data ($\mathcal{B}_K$), since this pollutes the quality of annotations on the new data, i.e., $\mathcal{L}_{Joint} = \mathcal{L}_{old}(\hat{\mathcal{B}}_M)$ and $\mathcal{L}_{LPL} = \mathcal{L}_{oldLPL}(\hat{\mathcal{B}}_M)$, and (2) we mask out gradients on pixel regions (around $0.3\%$ of all pixels) where the given V1.2 labels do not align with the parent class of their V2.0 labels for $\mathcal{L}_{old}(\hat{\mathcal{B}}_M)$ and $\mathcal{L}_{oldLPL}(\hat{\mathcal{B}}_M)$.

**Training and Inference Details**. For both **TrainRandom** and **TrainScratch**, we use an initial learning rate of 0.03; for **FinetunePrev**, we use an initial learning rate of 0.003. The L2 weight decay is selected as 0.0005. Other hyperparameters followed the default strategy in HRNet used to achieve the SOTA results on Cityscape. We use SGD with 0.9 momentum and a batch size of 16 for $\mathcal{B}_K$ (or 8 for $\mathcal{B}_K$ and 8 for $\hat{\mathcal{B}}_M$ if we use $\mathcal{L}_{SSL}$ or $\mathcal{L}_{Joint}$). We use linear learning rate decay schedule that sets the learning rate to $\eta(1 - \frac{k}{K})^{0.9}$ where $\eta$ is the initial learning rate and $k/K$ are the current and total iteration. For data augmentation, an input image and its label map are randomly flipped horizontally and then its longer edge will be scaled with a base size of 2200 pixels multiplied by a scalar factor sampled between 0.5 and 2.1. Finally, a 720x720 region will be randomly cropped for each of the 16 images in mini-batch for training. For inference, we use a sliding window of 720x720 without scaling of the image to determine the final pixel-level prediction. We also perform inference on the flipped image and take the average prediction as final result. Note that it is possible to use more advanced inference techniques such as multi-scaled testing, but we omit them to speed up inference since they are orthogonal to our research goal. We allocate the same training budget for all experiments. In particular, we use a total number of 1600 epochs for 2.5k/9k samples (a single TP), or 800 epochs for 5k/18k samples (two TPs data such as **AllFine** and **LabelNew** with SSL/Joint/LPL techniques). All experiments are conducted on internal clusters with 8 GeForce RTX 3090 cards.

## D: Dataset Statistics

To better understand the long-tailed nature of the datasets, we include the per-class image and pixel distributions for *iNat-LECO* in Figure 4 and *Mapillary-LECO* in Figure 5. We note that the long-tailed class distributions make semi-supervised methods less effective, as shown in Table 3, 4, 10, 11, e.g., the pseudo-labeling method of SSL underperforms the simple supervised learning baseline ($\mathcal{L}$ only).

Table 8: **Results of baseline methods for Mapillary-LECO**, analogous to Table 2 in the main paper. We report the mean Intersection-over-Union (mIoU in %). **TrainRandom** trains a model from scratch, as is the protocol in the main paper. Because segmentation tasks typically make use of ImageNet pre-training, we also show results for **TrainScratch** that trains with an ImageNet-pretrained encoder, improving upon **TrainRandom**). Finally, **FinetunePrev** makes use of TP[0]'s model, which is trained with **TrainScratch**. In Table 10, we explore Joint-Training and SSL, leading to further improvement.

| Benchmark | #images / TP | TP[1] Strategy | TP[1] mIoU | | |
|---|---|---|---|---|---|
| | | | LabelNew | RelabelOld | AllFine |
| *Mapillary-LECO* | 2500 | TrainRandom | 27.24 | 27.21 | 27.73 |
| | | TrainScratch | 28.22 | 29.28 | 30.71 |
| | | FinetunePrev | 30.39 | 29.05 | 31.73 |
| *Mapillary-LECO* | 9000 | TrainScratch | 32.83 | 33.44 | 33.82 |
| | | FinetunePrev | 34.08 | 35.01 | 36.20 |

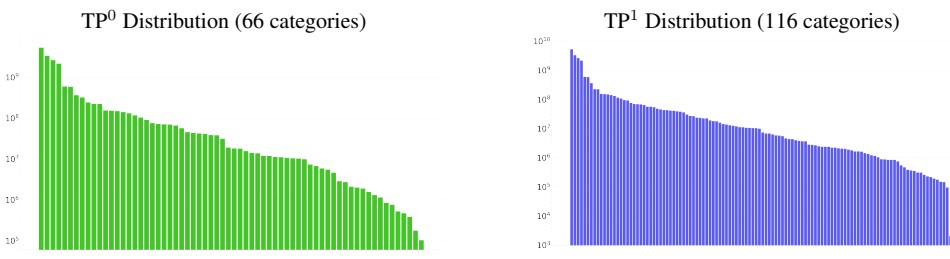

*Mapillary-LECO* Per-Class Pixel Distributions (Y-axis is log-scaled)

Figure 5: **Mapillary-LECO Per-Class Distributions.** We plot distributions of training pixels for *iNat-LECO* in log-scale. The x-axis are classes sorted by the number of pixels. The y-axis shows the number of pixels per class (in log scale).

## E: Training Details on CIFAR-LECO and iNat-LECO

We include all training details for reproducibility in this section. We adopt SOTA training strategies from recent papers [68–70]. For all experiments, we train the CNN with mini-batch stochastic gradient descent with 0.9 momentum. Same as FixMatch [68], we use the an exponential moving average (EMA) of model parameters for final inference with a decay parameter of 0.999. We use a cosine annealing learning rate decay schedule which sets the learning rate to $\eta \cos \frac{7\pi k}{16K}$ where $\eta$ is the initial learning rate and $k/K$ are the current and total iteration. By default, we use a strong data augmentation scheme because it provides better generalization performance. In particular, we adopt RandAugment [16] as implemented in this repository.[6] We did a grid search for best learning rate and weight decay using the validation set of each benchmark. The benchmark-specific hyperparameters are detailed below:

**CIFAR100-LECO**. We use the same model architecture (Wide-ResNet-28-2 [82]) in FixMatch paper for CIFAR100 experiments [68]. We use a batch size of 128 (if using $\mathcal{L}$ only), or we split the batch to 64 for $\mathcal{B}_K$ and 64 for $\hat{\mathcal{B}}_M$. For each experiment, we search for initial learning rate in $[0.6, 0.06, 0.006, 0.0006]$ and report the best mAcc result. We use an L2 weight decay of 5e-4. We run for a total number of 160K iterations and evaluate on the validation set per 1K iterations (equivalent to around 2000 epochs for 10000 images).

**iNat-LECO**. We use the same model architecture (ResNet50 [28]) as in [70]. We use a batch size of 60 (if using $\mathcal{L}$ only), or we split the batch to 30 for $\mathcal{B}_K$ and 30 for $\hat{\mathcal{B}}_M$. Because it is a long-tailed recognition problem, we search for L2 weight decay as suggested by [2] in $[0.001, 0.0001, 0.00001]$ and found out 0.001 produces the best mAcc. Then for each experiment, we search for initial learning rate in $[0.01, 0.001, 0.0001]$ and report the best mAcc result. We run for a total number of 250K iterations and evaluate on the validation set per 1K iterations (equivalent to around 300 epochs for 50000 images).

---

[6] https://github.com/kekmodel/FixMatch-pytorch/

Table 9: **Results of baseline methods for LECO** (complete results). For all our experiments except on Mapillary, we run each method five times and report the mean accuracy averaged over per-class accuracies with standard deviations. Because running on Mapillary is very compute-demanding, we run it once and report mean intersection-over-union (mIoU in %) over per-class IoU. For each benchmark, we bold the best mAcc/mIoU including ties that fall within its std. All conclusions we derived in main paper still hold, e.g., **FinetunePrev** is the best learning strategy that outperforms **TrainScratch** (cf. 73.64% vs. 65.69% with **LabelNew** on iNat-LECO), implying that the coarse-to-fine evolved ontology serves as a good learning curriculum. **FreezePrev** significantly underperforms **TrainScratch** (cf. 50.82% vs. 65.69% with LabelNew on iNat-LECO), confirming the difficulty of learning new classes that refine the old. **LabelNew** and **RelabelOld** achieve similar performance with the former using only new data but not the old altogether. If using both old and new data, we boost performance as shown in Table 10. The conclusions hold for varied number of training images and generalize to the Mapillary experiments, yet all methods fall short to the **AllFine** (which trains on all data assumed to be labeled with new ontology).

| Benchmark | #images / TP | $TP^0$ mAcc/mIoU | $TP^1$ Strategy | $TP^1$ mAcc/mIoU | | |
| --- | --- | --- | --- | --- | --- | --- |
| | | | | LabelNew | RelabelOld | AllFine |
| *CIFAR-LECO* | 1000 | $50.92 \pm 0.89$ | FinetunePrev | $\mathbf{33.53 \pm 0.57}$ | $32.85 \pm 0.47$ | $42.27 \pm 0.83$ |
| | | | FreezePrev | $24.13 \pm 0.67$ | $24.70 \pm 0.41$ | $27.82 \pm 0.43$ |
| | | | TrainScratch | $30.88 \pm 1.05$ | $31.02 \pm 0.68$ | $41.74 \pm 0.56$ |
| *CIFAR-LECO* | 2000 | $59.48 \pm 0.66$ | FinetunePrev | $\mathbf{43.44 \pm 0.73}$ | $\mathbf{42.88 \pm 0.72}$ | $52.72 \pm 0.38$ |
| | | | FreezePrev | $33.74 \pm 1.46$ | $33.69 \pm 1.44$ | $36.57 \pm 1.68$ |
| | | | TrainScratch | $41.61 \pm 0.53$ | $41.77 \pm 0.79$ | $52.78 \pm 0.35$ |
| *CIFAR-LECO* | 10000 | $77.78 \pm 0.27$ | FinetunePrev | $\mathbf{65.83 \pm 0.53}$ | $\mathbf{65.30 \pm 0.32}$ | $71.30 \pm 0.41$ |
| | | | FreezePrev | $52.64 \pm 0.50$ | $52.60 \pm 0.52$ | $53.79 \pm 0.43$ |
| | | | TrainScratch | $\mathbf{65.40 \pm 0.53}$ | $64.98 \pm 0.34$ | $71.43 \pm 0.24$ |
| *iNat-LECO* | 50000 | $64.21 \pm 1.61$ | FinetunePrev | $\mathbf{73.64 \pm 0.46}$ | $71.26 \pm 0.54$ | $84.51 \pm 0.66$ |
| | | | FreezePrev | $50.82 \pm 0.68$ | $54.60 \pm 0.49$ | $54.08 \pm 0.31$ |
| | | | TrainScratch | $65.69 \pm 0.46$ | $65.78 \pm 0.56$ | $73.10 \pm 0.80$ |
| *Mapillary-LECO* | 2500 | $37.01$ | FinetunePrev | $\mathbf{30.39}$ | $29.05$ | $31.73$ |
| | | | TrainScratch | $27.24$ | $27.21$ | $27.73$ |
| *Mapillary-LECO* | 9000 | $44.14$ | FinetunePrev | $34.08$ | $\mathbf{35.01}$ | $36.20$ |
| | | | TrainScratch | $32.83$ | $33.44$ | $33.82$ |

**iNat-4TP-LECO**. The range of grid search of hyperparameters follows that of **iNat-LECO**, including 250K iterations (equivalent to around 600 epochs for 25000 images) for each TP.

All the classification experiments were performed on a single modern GPU card. Based on the batch size scale, we use different GPU types, e.g., GeForce RTX 2080 Ti for image classification on CIFAR and iNaturalist, and GeForce RTX 3090 Ti on Mapillary.

## F: Comprehensive Experiment Results

We now show the complete results on varying sample sizes per TP for all benchmarks. Table 9, 10, 11 in appendix correspond to Table 2, 3, 4 in main paper. All conclusions generalize to varied number of samples and all 3 datasets.

We also have complete results using all combinations of the various SSL/Joint/LPL techniques introduced in main paper. Results can be found on Table 12 (*CIFAR-LECO* and *iNat-LECO*) and Table 13 (*Mapillary-LECO*).

Table 10: **Results of SSL and Joint Training methods** (complete results). Following the notation of Table 9, we study **LabelNew** annotation strategies that make use of algorithms for learning from old examples *without* exploiting the relationship between old-vs-new labels. Recall that AllFine uses $\mathcal{L}$ only to train on all the data labeled with new ontology (doubling annotation cost). Using all the three losses together consistently improves mAcc / mIoU for all the benchmarks, confirming the benefit of exploiting the old examples. For instance, on *iNat-LECO*, this boosts the performance to 83.6%, approaching the AllFine (84.5%)! Similar trends hold for *Mapillary-LECO*.

| Benchmark | #images / TP | SSL Alg | TP[1] mAcc/mIoU | | | | |
| --- | --- | --- | --- | --- | --- | --- | --- |
| | | | $\mathcal{L}$ only | $+\mathcal{L}_{SSL}$ | $+\mathcal{L}_{Joint}$ | $+\mathcal{L}_{Joint}$ $+\mathcal{L}_{SSL}$ | AllFine |
| *CIFAR-LECO* | 1000 | ST-Hard | $33.53 \pm 0.57$ | $34.59 \pm 0.57$ | $37.25 \pm 0.66$ | $35.27 \pm 0.64$ | $42.27 \pm 0.83$ |
| | | ST-Soft | | $37.67 \pm 0.46$ | | $\mathbf{38.48 \pm 0.67}$ | |
| | | PL | | $33.53 \pm 0.36$ | | $37.28 \pm 0.67$ | |
| | | FixMatch | | $34.20 \pm 0.24$ | | $37.47 \pm 0.53$ | |
| *CIFAR-LECO* | 2000 | ST-Hard | $43.44 \pm 0.73$ | $44.87 \pm 0.35$ | $47.68 \pm 0.70$ | $45.67 \pm 0.32$ | $52.72 \pm 0.38$ |
| | | ST-Soft | | $\mathbf{48.23 \pm 0.55}$ | | $\mathbf{48.72 \pm 0.53}$ | |
| | | PL | | $43.46 \pm 0.61$ | | $47.24 \pm 0.40$ | |
| | | FixMatch | | $44.98 \pm 0.59$ | | $\mathbf{48.58 \pm 0.71}$ | |
| *CIFAR-LECO* | 10000 | ST-Hard | $65.83 \pm 0.27$ | $67.11 \pm 0.34$ | $68.74 \pm 0.21$ | $67.97 \pm 0.37$ | $71.30 \pm 0.41$ |
| | | ST-Soft | | $\mathbf{69.86 \pm 0.30}$ | | $\mathbf{69.77 \pm 0.71}$ | |
| | | PL | | $65.86 \pm 0.14$ | | $68.43 \pm 0.40$ | |
| | | FixMatch | | $67.13 \pm 0.36$ | | $69.03 \pm 0.43$ | |
| *iNat-LECO* | 50000 | ST-Hard | $73.64 \pm 0.46$ | $75.23 \pm 0.14$ | $82.98 \pm 0.33$ | $78.23 \pm 0.38$ | $84.51 \pm 0.66$ |
| | | ST-Soft | | $79.64 \pm 0.41$ | | $\mathbf{83.61 \pm 0.39}$ | |
| | | PL | | $73.56 \pm 0.09$ | | $82.87 \pm 0.51$ | |
| | | FixMatch | | $74.57 \pm 0.54$ | | $83.00 \pm 0.33$ | |
| *Mapillary-LECO* | 2500 | ST-Hard | $30.39$ | $30.06$ | $\mathbf{31.05}$ | $\mathbf{31.09}$ | $31.73$ |
| *Mapillary-LECO* | 9000 | ST-Hard | $34.08$ | $32.68$ | $\mathbf{35.40}$ | $\mathbf{35.72}$ | $36.20$ |

Table 11: **Results of exploiting taxonomic hierarchy** (complete results). We copy the best results in Table 10 (all obtained by $+\mathcal{L}_{Joint}$ $+\mathcal{L}_{SSL}$) for reference. We can see that $\mathcal{L}_{LPL}$ is less effective but still competitive on *Mapillary-LECO*, presumably because it does not model the class-merging scenarios. Nonetheless, combinations of the strategies, i.e., adding SSL loss along with pseudo-label refinement via either **Filtering** or **Conditioning** still bridge the gap to AllFine.

| Benchmark | #images / TP | SSL Alg | TP[1] mAcc | | | | |
| --- | --- | --- | --- | --- | --- | --- | --- |
| | | | $+\mathcal{L}_{Joint}$ $+\mathcal{L}_{SSL}$ | $+\mathcal{L}_{LPL}$ | $+\mathcal{L}_{LPL}$ $+\mathcal{L}_{SSL/Filter}$ | $+\mathcal{L}_{LPL}$ $+\mathcal{L}_{SSL/Cond}$ | AllFine |
| *CIFAR-LECO* | 1000 | ST-Hard | $38.48 \pm 0.67$ | $38.47 \pm 0.52$ | $\mathbf{38.87 \pm 0.54}$ | $\mathbf{38.68 \pm 0.41}$ | $42.27 \pm 0.83$ |
| | | ST-Soft | | | $\mathbf{38.87 \pm 0.52}$ | $38.19 \pm 0.71$ | |
| | | PL | | | $\mathbf{38.56 \pm 0.44}$ | $\mathbf{38.67 \pm 0.81}$ | |
| | | FixMatch | | | $37.55 \pm 0.69$ | $37.35 \pm 0.72$ | |
| *CIFAR-LECO* | 2000 | ST-Hard | $48.72 \pm 0.53$ | $49.04 \pm 0.47$ | $\mathbf{49.36 \pm 0.56}$ | $48.65 \pm 0.53$ | $52.72 \pm 0.38$ |
| | | ST-Soft | | | $49.09 \pm 0.46$ | $48.54 \pm 0.52$ | |
| | | PL | | | $\mathbf{49.25 \pm 0.34}$ | $48.91 \pm 0.87$ | |
| | | FixMatch | | | $\mathbf{49.44 \pm 0.22}$ | $48.81 \pm 0.40$ | |
| *CIFAR-LECO* | 10000 | ST-Hard | $69.77 \pm 0.71$ | $69.31 \pm 0.30$ | $69.90 \pm 0.29$ | $\mathbf{70.42 \pm 0.23}$ | $71.30 \pm 0.41$ |
| | | ST-Soft | | | $70.10 \pm 0.12$ | $70.02 \pm 0.63$ | |
| | | PL | | | $69.20 \pm 0.20$ | $69.60 \pm 0.26$ | |
| | | FixMatch | | | $69.61 \pm 0.17$ | $69.85 \pm 0.46$ | |
| *iNat-LECO* | 50000 | ST-Hard | $83.61 \pm 0.39$ | $83.62 \pm 0.21$ | $\mathbf{84.23 \pm 0.61}$ | $\mathbf{84.34 \pm 0.25}$ | $84.51 \pm 0.66$ |
| | | ST-Soft | | | $\mathbf{84.12 \pm 0.31}$ | $83.76 \pm 0.40$ | |
| | | PL | | | $83.72 \pm 0.22$ | $\mathbf{84.16 \pm 0.19}$ | |
| | | FixMatch | | | $83.91 \pm 0.39$ | $83.95 \pm 0.40$ | |
| *Mapillary-LECO* | 2500 | ST-Hard | $31.09$ | $31.04$ | $\mathbf{31.41}$ | $31.23$ | $31.73$ |
| *Mapillary-LECO* | 9000 | ST-Hard | $35.72$ | $35.04$ | $\mathbf{36.12}$ | $35.47$ | $36.20$ |

Table 12: **Complete results of LECO strategies on classification benchmarks**. We report results on combinations of all LECO stratgies on *CIFAR100-LECO* and *iNat-LECO* with **FinetunePrev** strategy on $TP^1$. Major conclusions hold across all benchmarks: (1) Utilizing old-ontology samples and labels, i.e., $\mathcal{L}_{SSL}$ and $\mathcal{L}_{Joint}$, helps bridge the gap to **AllFine**, and (2) exploiting taxonomic hierarchy via $\mathcal{L}_{SSL/Filter}$, $\mathcal{L}_{SSL/Cond}$, or $\mathcal{L}_{LPL}$ further improves the performance.

| Setup | $TP^0$ mAcc | AllFine | No SSL $\mathcal{L}$ | $\mathcal{L}_{LPL}$ | $\mathcal{L}_{Joint}$ | | PL $\mathcal{L}$ | $\mathcal{L}_{LPL}$ | $\mathcal{L}_{Joint}$ | FixMatch $\mathcal{L}$ | $\mathcal{L}_{LPL}$ | $\mathcal{L}_{Joint}$ | ST-Hard $\mathcal{L}$ | $\mathcal{L}_{LPL}$ | $\mathcal{L}_{Joint}$ | ST-Soft $\mathcal{L}$ | $\mathcal{L}_{LPL}$ | $\mathcal{L}_{Joint}$ |
|---|---|---|---|---|---|---|---|---|---|---|---|---|---|---|---|---|---|---|
| CIFAR100-1k | 50.92±0.89 | 42.27±0.83 | 33.53±0.57 | 38.47±0.52 | 37.25±0.66 | $+\mathcal{L}_{SSL}$ | 33.53±0.36 | 38.70±0.65 | 37.28±0.67 | 34.20±0.24 | 37.56±0.54 | 37.47±0.53 | 34.59±0.57 | 36.78±0.40 | 35.27±0.64 | 37.67±0.46 | 38.80±0.49 | 38.48±0.67 |
| | | | | | | $+\mathcal{L}_{SSL/Filter}$ | 33.33±0.69 | 38.56±0.44 | 37.44±0.84 | 36.02±0.33 | 37.55±0.69 | 37.23±0.20 | 37.93±0.50 | 38.87±0.54 | 38.57±0.74 | 37.59±0.69 | 38.87±0.52 | 38.43±0.67 |
| | | | | | | $+\mathcal{L}_{SSL/Cond}$ | 37.40±1.19 | 38.67±0.81 | 38.00±0.59 | 36.92±0.64 | 37.35±0.72 | 37.08±0.57 | 37.95±0.34 | 38.68±0.41 | 38.20±0.30 | 38.25±0.62 | 38.19±0.71 | 38.45±0.49 |
| CIFAR100-2k | 59.48±0.66 | 52.72±0.38 | 43.44±0.73 | 49.04±0.47 | 47.68±0.70 | $\mathcal{L}_{SSL}$ | 43.46±0.61 | 48.82±0.30 | 47.24±0.40 | 44.98±0.59 | 49.38±0.29 | 48.58±0.71 | 44.87±0.35 | 46.98±0.33 | 45.67±0.32 | 48.23±0.55 | 49.12±0.50 | 48.72±0.53 |
| | | | | | | $\mathcal{L}_{SSL/Filter}$ | 43.50±0.47 | 49.25±0.34 | 47.38±0.60 | 46.14±0.33 | 49.44±0.22 | 48.23±1.21 | 47.00±0.36 | 49.36±0.56 | 49.01±0.73 | 47.62±0.74 | 49.09±0.46 | 48.56±0.50 |
| | | | | | | $\mathcal{L}_{SSL/Cond}$ | 48.28±0.27 | 48.91±0.87 | 48.45±0.34 | 48.23±0.93 | 48.81±0.40 | 48.57±0.58 | 48.81±0.57 | 48.65±0.53 | 48.60±0.68 | 48.70±0.58 | 48.54±0.52 | 48.86±0.61 |
| CIFAR100-10k | 77.78±0.27 | 71.30±0.41 | 65.83±0.27 | 69.31±0.30 | 68.74±0.21 | $\mathcal{L}_{SSL}$ | 65.86±0.14 | 69.48±0.24 | 68.43±0.40 | 67.13±0.36 | 69.71±0.49 | 69.03±0.43 | 67.11±0.34 | 68.58±0.51 | 67.97±0.37 | 69.86±0.30 | 68.96±0.27 | 69.77±0.71 |
| | | | | | | $\mathcal{L}_{SSL/Filter}$ | 65.80±0.34 | 69.20±0.20 | 68.82±0.23 | 67.60±0.25 | 69.61±0.17 | 69.15±0.41 | 69.04±0.29 | 69.90±0.29 | 69.95±0.28 | 68.76±0.20 | 70.10±0.12 | 69.85±0.38 |
| | | | | | | $\mathcal{L}_{SSL/Cond}$ | 68.32±0.26 | 69.60±0.26 | 69.09±0.13 | 68.86±0.52 | 69.85±0.46 | 69.41±0.39 | 70.22±0.38 | 70.42±0.23 | 70.05±0.18 | 70.33±0.15 | 70.02±0.63 | 69.93±0.24 |
| iNat-50k | 64.21±1.61 | 84.51±0.66 | 73.64±0.46 | 83.62±0.21 | 82.98±0.33 | $\mathcal{L}_{SSL}$ | 73.56±0.09 | 83.71±0.44 | 82.87±0.51 | 74.57±0.54 | 84.05±0.26 | 83.00±0.33 | 75.23±0.14 | 79.07±0.48 | 78.23±0.38 | 79.64±0.41 | 84.10±0.54 | 83.61±0.39 |
| | | | | | | $\mathcal{L}_{SSL/Filter}$ | 74.01±0.41 | 83.72±0.22 | 82.81±0.27 | 74.55±0.33 | 83.91±0.39 | 83.16±0.33 | 79.91±0.58 | 84.23±0.61 | 83.92±0.25 | 78.71±0.22 | 84.12±0.31 | 83.80±0.60 |
| | | | | | | $\mathcal{L}_{SSL/Cond}$ | 81.98±0.12 | 84.16±0.19 | 82.99±0.64 | 82.62±0.21 | 83.95±0.40 | 83.66±0.36 | 84.20±0.29 | 84.34±0.25 | 84.37±0.39 | 83.96±0.47 | 83.76±0.40 | 83.95±0.20 |

Table 13: **Complete results of LECO strategies on** *Mapillary-LECO*. We report results on combinations of all LECO stratgies on *Mapillary-LECO* with **FinetunePrev** strategy on $TP^1$. It is worth noting that new labels in V2.0 do not necessarily have surjective mappings to old labels of V1.2. This means, it is non-trivial to apply coarse supervision as partial labels. Moreover, we only perform **ST-Hard** for SSL because other methods are excessively compute-demanding (e.g., ST-SOft requires saving per-class heatmaps for each image to allow making use of pseudo-labels' scores, FixMatch requires learning a separate large-scale model, etc.). Concretely, we (1) do not add coarse supervision for $\mathcal{B}_K$ since it pollutes quality of annotation on new data, and (2) mask out gradient where the given V1.2 labels do not align with the parent class for $\mathcal{L}_{old}(\hat{\mathcal{B}}_M)$ and $\mathcal{L}_{oldLPL}(\hat{\mathcal{B}}_M)$. Major conclusions hold here. We can see that all combinations of various LECO stratgies achieve suprisingly good performances, whereas the best combination bridges 95% of the gap to **AllFine** (gap reduces from 1.34 to 0.08 for 2.5k samples with $\mathcal{L}_{SSL}$ and $\mathcal{L}_{LPL}$, and from 2.12 to 0.08 for 9k samples with $\mathcal{L}_{SSL/Filter}$ and $\mathcal{L}_{LPL}$).

| Setup | $TP^0$ mIoU | AllFine | No SSL $\mathcal{L}$ | $\mathcal{L}_{LPL}$ | $\mathcal{L}_{Joint}$ | | ST-Hard $\mathcal{L}$ | $\mathcal{L}_{LPL}$ | $\mathcal{L}_{Joint}$ |
|---|---|---|---|---|---|---|---|---|---|
| Mapillary-2.5k | 37.01 | 31.73 | 30.39 | 31.04 | 31.05 | $+\mathcal{L}_{SSL}$ | 30.06 | 31.65 | 31.09 |
| | | | | | | $+\mathcal{L}_{SSL/Filter}$ | 30.64 | 31.41 | 31.57 |
| | | | | | | $+\mathcal{L}_{SSL/Cond}$ | 31.01 | 31.23 | 31.50 |
| Mapillary-9k | 44.14 | 36.20 | 34.08 | 35.04 | 35.40 | $+\mathcal{L}_{SSL}$ | 32.68 | 35.97 | 35.72 |
| | | | | | | $+\mathcal{L}_{SSL/Filter}$ | 34.96 | 36.12 | 36.06 |
| | | | | | | $+\mathcal{L}_{SSL/Cond}$ | 35.29 | 35.47 | 35.70 |