# OpenReview forum: "Continual Learning with Evolving Class Ontologies"
_NeurIPS.cc/2022/Conference — NeurIPS 2022 Accept_

### Official Review · Reviewer_kuwA · 2022-07-10

**Rating:** 6
**Confidence:** 3
**Soundness:** 3 good
**Presentation:** 3 good
**Contribution:** 3 good

**Summary:**

The paper introduces a new continual learning problem setup, where class vocabulary becomes more fine grained in a continual fashion. Different than classic continual learning this setup allows access to the historical examples. Thus it is not prone to catastrophic forgetting.

The paper explores several research questions, like when the vocabulary evolves whether to annotate new data or relabel the old data (without collecting new data), how to leverage coarse label (of old data), and whether to finetune the model trained on old data, or train from scratch.

The paper show that a semi supervised approach, that only requires labelling the new data without relabeling the old data, is almost equivalent to relabeling all the data (both old and new).

The approach uses the new data to provide pseudo labels for the old data, and use the old data coarse labels to resolve reconcile conflicts between the pseudo fine labels and the true coarse labels.


**Questions:**


1. It is not clear in what the new problem setup is different than LPL


2. The approach is somewhat similar to [4], which was applied to a different problem setup. I suggest citing it in the related work.

###References
[4] Goel et al, Not All Relations are Equal: Mining Informative Labels for Scene Graph Generation, CVPR 2022

**Limitations:**

Ok,

See my feedback in the weakness section


**Strengths And Weaknesses:**


### Strengths

* The paper is well written, very easy to read, and provides decent baselines and upper bounds.

* The problem setup is important from a practical standpoint, and novel to the best of my knowledge (I am not an expert in continual learning, or hierarchical learning).

* The suggested approach saves relabeling efforts, which can become quadratic with the number of "time periods".


### Weaknesses

* Although the problem setup is general, and allowing multiple "time periods" (TP), in practice the experiment are only with a single TP. I think this is the major weakness of the paper, because it is not clear how well the approach generalizes, as the labels become more fine grained.

* The approach was only evaluated on two datasets. It would be useful to provide experiments for a different data domain (not vision).

* The problem setup ignores the fact that aside from storage, retraining on the old samples may be very costly from a compute standpoint.  E.g. retraining a model with a scale of Giga samples like a in autonomous driving may cost millions of dollars. There are recent works like [1,2,3] that try to alleviate this problem. I suggest to discuss it in the related work.

* This setup may introduce biases. There may be a distribution shift when collecting more fine grained labels. It would have been beneficial to add this type of bias to the benchmark, and demonstrate how sensitive is the approach to such a shift.

* It is unclear what are the open questions and the next steps. If the approach reaches the upper bound, does it mean that this problem is solved?

* Experiments: There is a better upper bound. Which is utilizing both the old and new labels for the old data


### References
[1] Houlsby et al. Parameter-Efficient Transfer Learning for NLP, ICML 2019

[2] Li et al, Cross-domain Few-shot Learning with Task-specific Adapters, CVPR 2022

[3] Cohen et al, "This is my unicorn, Fluffy": Personalizing frozen vision-language representations, ECCV 2022

---

> ### Author Response · Authors · 2022-08-01
> **Rebuttal to Reviewer kuwA (1/2)**
>
> We thank Reviewer kuwA for the valuable comments and appreciating our well-written paper, important and novel problem setup, and effective approaches. Along with the rebuttal, **we have updated our paper by including the experiments on the large-scale Mapillary dataset**, which was included in the supplement. The Mapillary dataset was collected for semantic segmentation research in the context of autonomous driving; its classes follow a long-tailed class distribution. Importantly, it reflects **a real-world case of LECO** because its ontologies were versioned from V1.2 to V2.0, formulated as TP$^0$ and TP$^1$ in our experiments. Below, we address specific issues with referred line numbers in the updated paper.
>
> > **Reviewer kuwA thinks a major weakness of the paper is that experiments set up only two time periods (TPs), and is afraid that the conclusions might not generalize to more TPs.**
>
> We see the worry as a result of relating LECO with continual learning (CL), which is a different problem. CL emphasizes the issue of catastrophic forgetting, and proposes to exacerbate this issue by setting more TPs (and using a small buffer for storing training data). Differently, LECO emphasizes the difficulty of learning for new classes which are subclasses of the previously learned (super)classes (Line95), without necessarily restricting a buffer size. Results in Table 2 show that, even without buffer restrictions, CL methods still perform poorly (cf. FreezePrev [47] and TrainScratch [59]). Therefore, using two TPs sufficiently emphasizes the difficulty in LECO. We added an explanation for this concern in the updated paper (Line147-152).
>
> Furthermore, our explored techniques are not affected by the number of TPs. Because more TPs provide more data, our techniques will improve further by exploiting such data, e.g., using techniques of semi-supervised learning, joint training, and learning with partial labels. Moreover, we note that using two time periods is also the more common problem formulation in the context of dataset versioning (e.g., Mapillary and Argoverse, cf. Line36-38), implying a solution for it itself may already have widespread impact.
>
> > **Reviewer kuwA is concerned that our experiments only have two datasets.**
>
> The updated paper contains experiments on the third dataset Mapillary (cf. Mapillary-LECO in Table 1-4), which is large-scale, originally collected for semantic segmentation in the context of autonomous driving research. These experiments were included in our supplement. The Mapillary dataset was versioned from V1.2 to V2.0, reflecting a real-world case where labels evolved from coarse to fine.
>
> > **Reviewer kuwA points out that retraining "may be very costly from a compute standpoint", and recommends some recent papers that propose to improve model adaptation or finetuning.**
>
> Thank you for the recommended papers; we cited them in the updated paper (Line334). We agree that retraining on large-scale data can be costly and non-trivial. Perhaps fortunately, our explorations demonstrate that a better strategy is finetuning, instead of retraining (Table 2 and Line331). That said, techniques that aim for efficient adaptation/finetuning can be readily applied to LECO.
>
> > **Reviewer kuwA points out that data distribution shift is interesting to explore.**
>
> Agreed! We have noticed this in Line335 and plan to set up this scenario in future work.
>
> > **Reviewer kuwA asks about next steps based on this work, and concerned that LECO might be solved because our explored method approaches the upper-bound.**
>
> As discussed in Line325-336, next steps include a new setup that incorporates unlabeled data, shifting/dynamic data distributions, etc. We believe such new LECO setups have better upper bounds and offer new exploration space.
>
> > **Reviewer kuwA suggests a better upper bound which is to use all levels of labels and both old and new data.**
>
> Great suggestion! We will update the upper bounds after we finish training the upperbound models. The training is straightforward but takes about one month on all datasets (particularly the large-scale ones on iNaturalist and Mapillary).

---

> > ### Author Response · Authors · 2022-08-01
> > **Rebuttal to Reviewer kuwA (2/2)**
> >
> > > **Reviewer kuwA asks about the difference between LECO and learning with partial labels (LPL).**
> >
> > LECO studies how to learn models for new classes which are subclasses of previously learned (super)classes. The foremost question to answer is how to annotate data with new classes (Line55), i.e., relabeling the old or labeling the new. Our experiments convincingly demonstrate that labeling the new is a much better strategy, hence leading to the second question about how to leverage coarsely-labeled old data and fine-labeled new data. While LPL methods can be readily adopted, LECO offers the opportunity to study whether/how to exploit previous TP's model. Our extensive experiments (Table 2) show finetuning significantly outperforms training from scratch, which is the setup for LPL. That said, LECO is a richer venue to explore algorithms than the setup for LPL.
> >
> > > **Reviewer kuwA recommended a paper that studies how to mine label relations for scene graph generation using semi-supervised learning.**
> >
> > Thank you for the recommended paper. We happily cited this paper (reference [25]) in our updated manuscript (Line221).

---

> > ### Comment · Reviewer_kuwA · 2022-08-07
> > **Two TPs evaluation**
> >
> > Thank you for your answers. I still find the two TPs evaluation as the main weakness of the paper. Although the rebuttal suggests that the approach would generalize to more TPs, there is no evidence that demonstrate this claim. Therefore I keep my evaluation as is.
> >
> > In general i disagree with the borderline reject scores of the other reviews and suggest weak accept.

---

> > > ### Author Response · Authors · 2022-08-08
> > > **thank you; we'll add an experiment with more TPs in camera-ready**
> > >
> > > Thank you Reviewer kuwA for your interaction and feedback!
> > >
> > > As for why our approach generalizes to more TPs, besides the explanations in our rebuttal, we agree that an experiment with more TPs will be a good empirical demonstration. Therefore, we will happily set up such an experiment with four TPs on iNaturalist, where we define the four ontologies based on four of its taxa levels (i.e., "order", "family", "genus", and "species"). Our open-source codebase (cf. supplemental material) makes completing this experiment straightforward. We will add it in the camera-ready version. We are open to further suggestions about this experiment.

---

### Official Review · Reviewer_YxYF · 2022-07-11

**Rating:** 5
**Confidence:** 3
**Soundness:** 2 fair
**Presentation:** 3 good
**Contribution:** 1 poor

**Summary:**

The paper introduces the problem of Learning with Evolving Class Ontology (LECO), a general case of Class Incremental Learning, where instead of introducing new classes at each time step, we refine existing class labels into more granular ones. The authors ask whether, given incoming data and a fixed annotation budget, it is better to annotate the new data with the new labels or re-annotate the old data. In other words, whether it is better to train a classifier on a smaller but homogenous dataset or a larger one with a mix of old (coarse) and new (fine) labels. Based on experiments on CIFAR and iNaturalist, the authors conclude that by using semi-supervised learning and incorporating the hierarchical taxonomy, it is possible to achieve excellent classification accuracy with a heterogeneous dataset. In particular, by using pseudo-labelling and Learning-with-Partial-Labels, it is possible to get close to the performance of a classifier trained on a homogenous dataset of the same size.

**Questions:**

In the "Technical Insights" section of the introduction, we read that "Under the LECO protocol of the iNaturalist benchmark (Section 3), (1) labelling new data outperforms relabeling the old: 73.3% vs 70.9% in accuracy;". I couldn't find these numbers in the result tables, so it is unclear how the label mismatch is handled in this base case. Is the strategy to learn a feature extractor on the old data and fine-tune it on the new data?

In the "Benchmarks" section of Chapter 3, the authors claim that "using all the labelled data is realistic because, for example, safety-critical systems (e.g., autonomous vehicles) must not trade off recognition accuracy with parsimonious memory buffer." While the buffer size is indeed arbitrary, for any practical continual learning system, it has to be *finite*. If we extend the LECO setting to multiple TPs, the buffer size will eventually become an important consideration.

One of the strategies in Chapter 4 is to "train an entire network from scratch with random weights." The phrasing is unclear here. Does it mean "train a randomly initialised neural network"?

Another strategy in Chapter 4 is to train a classifier on top of a randomly initialised feature extractor. What is the purpose of this? Is it to provide a lower bound on the classifier performance?

In Section 6.2., the first strategy is described as follows: "we filter out pseudo-labels that do not align with the ground-truth coarse labels." What does filter out mean in this context? Is the data point rejected? Is the coarse label used instead?

**Limitations:**

The authors correctly identified an essential caveat in their research question: obtaining new data is generally equally or more costly than annotating it. Re-annotation is also usually quicker than annotating from scratch because the annotator can use existing labels to constrain the task to a handful of classes.

One significant problem mentioned but not explored enough is the assumption that the findings would generalise to multiple TPs. In particular, it would be interesting to see the performance of Learning-with-Partial-Labels with multiple levels of granularity.

**Strengths And Weaknesses:**

The main strength of the paper is clarity. The structure is good, the writing is easy to understand, and the results are presented in a logical, incremental order. The authors precisely state the research questions and contributions and briefly summarise the conclusions in the introduction, which helps to guide the reader. The paper includes relevant details to reproduce the experiments.

I have no reservations about the quality of the submission. The experimental setup is sound, and the proposed models seem appropriate to tackle the presented problem. The authors evaluate their methods on two different datasets, modified to match the evolving ontology scenario. As mentioned below, the evaluation could be improved by including related work.

The biggest weakness of the submitted work is originality. While the LECO setting appears novel and could be an interesting type of class-incremental learning, limiting it to two time steps and giving the model access to all the data turns it into fine classification with coarse supervision, a problem tackled by other omitted work, such as:

- A Weakly Supervised Fine Label Classifier Enhanced by Coarse Supervision, Taherkhani et al. 2019
- Weakly Supervised Image Classification with Coarse and Fine Labels, Lei et al. 2017
- A Pseudo-Label Method for Coarse-to-Fine Multi-Label Learning with Limited Supervision, Hsieh et al. 2019
- From Categories to Subcategories: Large-scale Image Classification with Partial Class Label Refinement

The authors combine existing methods in a novel way, which could be a valuable contribution. However, the lack of comparison to previous approaches makes it hard to judge whether their work advances state of the art and therefore undermines the significance of their findings. My suggestion would be to either extend the work to multiple TPs or focus on the problem of fine classification with coarse supervision: remove the LECO formulation and compare their method with existing work. I think either would make the submission much stronger.

---

> ### Author Response · Authors · 2022-08-01
> **Rebuttal to Reviewer YxYF (1/2)**
>
> We thank Reviewer YxYF for the insightful comments and having "no reservation" about the paper quality (e.g., clarity and well-organized paper structure, the sound experimental setup and proposed models, etc.). Along with the rebuttal, **we have updated our paper by including the experiments on the large-scale Mapillary dataset**, which was included in the supplement. The Mapillary dataset was collected for semantic segmentation research in the context of autonomous driving; its classes follow a long-tailed class distribution. Importantly, **it reflects a real-world case of LECO** because its ontologies were versioned from V1.2 to V2.0, formulated as TP$^0$ and TP$^1$ in our experiment. Below, we address specific issues with referred line numbers in the updated paper.
>
> > **Reviewer YxYF is worried that the current LECO setup has two time periods (TPs).**
>
> We see the worry as a result of relating LECO with continual learning (CL), which is a different problem. CL emphasizes the issue of catastrophic forgetting, and proposes to exacerbate this issue by setting more TPs (and using a small buffer for storing training data). Differently, LECO emphasizes the difficulty of learning for new classes which are subclasses of the previously learned (super)classes (Line95), without necessarily restricting a buffer size. Results in Table 2 show that, even without buffer restrictions, CL methods still perform poorly (cf. FreezePrev [47] and TrainScratch [59]). Therefore, using two TPs sufficiently emphasizes the difficulty in LECO. We added an explanation for this concern in the updated paper (Line147-152).
>
> Furthermore, our explored techniques are not affected by the number of TPs. Because more TPs provide more data, our techniques will improve further by exploiting such data, e.g., using techniques of semi-supervised learning, joint training, and learning with partial labels. Moreover, we note that using two time periods is also the more common problem formulation in the context of dataset versioning (e.g., Mapillary and Argoverse, cf. Line36-38), implying a solution for it itself may already have widespread impact.
>
> > **Reviewer YxYF thinks the current LECO setup makes it similar to the problem of Fine-Grained Classification with both Fine and Coarse Supervision (FGC-FCS); it misses some citations and comparisons to the state-of-the-art methods in this line of work.**
>
> Thank you for recommending the papers which are among the first that studied the problem of FGC-FCS. We happily cited them (Line110). Importantly, for FGC-FCS, we cited and compared the methods introduced by Su et al. [69,70], which were published in 2021 and are the state-of-the-art (cf. Line221, 230-240). We point out that some of our most effective solutions (such as joint training) are general enough to apply for more complex ontology evolutions (such as those present in Mapillary).
>
> > **Reviewer YxYF points out typos in the referred accuracies in Introduction, and unclear statement how labeling-new-data and relabeling-the-old achieve different results in Table 2; asks whether it is because of learning a feature extractor on the old data and fine-tune it on the new data?**
>
> Thank you for pointing out the typos; we apologize! We have corrected them (Line79-83). Yes, you are right! We updated the sentence to make it clear (Line79): "finetuning the previous TP's model on newly-labeled-data outperforms that on relabeled-old-data: 73.64% vs. 71.26% in accuracy".
>
> > **Reviewer YxYF points out that using all the labeled data is unrealistic because buffer size has to be finite in practical continual learning systems.**
>
> It is right that the buffer size may become an issue for practical continual learning systems such as autonomous vehicles, which continuously update their machine-learned models and accumulate huge amounts of data over years. However, to the best of our knowledge, such systems do not have issues to store *labeled data*, though they may have issues for storing *unlabeled* data. For example, in [R1], interviewed auto-manufacturers argue for saving all (important) data for potential legal action. In fact, many other applications save history data for privacy-related consideration (e.g., medical data records) and as forensic evidence (e.g., videos from surveillance cameras). We hope the reviewer considers these real-world applications as realistic setups that use a buffer large enough to store all *labeled* data.
>
> [R1] Amend, J. M. (2018, January 18). Storage almost full: Driverless cars create data crunch. Wards Auto. Retrieved from https://www.wardsauto.com/technology/storage-almost-full-driverless-cars-create-data-crunch

---

> > ### Author Response · Authors · 2022-08-01
> > **Rebuttal to Reviewer YxYF (2/2)**
> >
> > > **Reviewer YxYF asks if "training an entire network from scratch with random weights" means "training a randomly initialised neural Network"?**
> >
> > Yes!
> >
> > > **Reviewer YxYF asks if "training a classifier on top of a randomly initialised feature extractor" serves as a lower-bound?**
> >
> > Yes! We removed this lower bound in the updated paper because it does not provide much meaningful analysis.
> >
> > > **Reviewer YxYF asks (1) if "filter out" means "rejecting data" in the sentence "we filter out pseudo-labels that do not align with the ground-truth coarse labels", and (2) if the coarse label is used?**
> >
> > Yes (Line296), and yes (Line295)!

---

> > > ### Comment · Reviewer_YxYF · 2022-08-08
> > > **Reply to the authors**
> > >
> > > Thank you for your response! Given a promised extension of the method to multiple TPs, I have upgraded my rating to borderline accept.

---

> > > > ### Author Response · Authors · 2022-08-09
> > > > **Thank you for your feedback and upgraded rating!**
> > > >
> > > > Thank you Reviewer YxYF for your interaction and upgraded rating!
> > > >
> > > > Again, we appreciate your positive attitude with "*no reservation about the quality of the submission*" (e.g., clarity and well-organized paper structure, the sound experimental setup and proposed models,  novel setting of LECO, novel approach that combines existing techniques, etc.). To strengthen our work, as promised, we will add the new experiment in the camera-ready (cf. setup below for reference). In the meantime, we are delighted to address any other concerns that hold you off upgrading further (e.g., your rating on contribution is "1 poor"; a typo?).
> > > >
> > > > *Setup for the new experiment with more TPs.* We repurpose iNaturalist to define four TPs, with each having an ontology out of four at distinct taxa levels ["order", "family", "genus", "species"]. We will use our open-source codebase (cf. supplemental material) to readily complete this experiment.

---

### Official Review · Reviewer_6YbD · 2022-07-11

**Rating:** 5
**Confidence:** 4
**Soundness:** 3 good
**Presentation:** 3 good
**Contribution:** 2 fair

**Summary:**

This paper introduces a new problem called Learning with Evolving Class Ontology (LECO) where the objective is to learn finer classes over time. The authors have shown that labeling new data with fine-grained annotation is more valuable and proposed to use the techniques from semi-supervised learning and learning with partial labels literature to utilize the old coarsely-grained data. The authors have proposed a benchmarking protocol for LECO on top of two image classification datasets: CIFAR100 and iNaturalist and shown promising results.

**Questions:**

- How were the levels selected for the iNaturalist-LECO experimental setup?

- The proposed method uses a large validation set. Is the final performance sensitive to the validation set size? Using such a large validation set to search hyperparameters is not very realistic. Were the upper-bound hyperparameters searched in a similar manner?

- What happens if the fine-grained splitting of a corase-grained class is long-tailed. Most of the pseudo-labels for the coarse-grained old class will be assigned to the head fine-grained child class.


**Limitations:**

The authors have adequately addressed the limitations and potential negative societal impact of their work.

**Strengths And Weaknesses:**

Strengths:

- Tackles a somewhat new and interesting problem

- The proposed solution is simple and intuitive

- Experiments are extensive and systematic

Weaknesses:

- The technical novelty of the proposed solution is low. This is not a bad thing in general. However, in this particular case, since the proposed problem (LECO) is somewhat similar to the existing continual learning problems and the proposed solution is an ensemble of multiple existing ideas I find this to be a major weakness.

- I find the current problem setup and solution a bit restrictive. It overlooks a lot of practical concerns. (a) assumes a negligible cost for data acquisition, if new data is not available one can only relabel old data. (b) only expects finer labels in subsequent time periods. What happens if a new category is introduced which doesn't have a parent. The pseudo-labeling approach will mistakenly assign previous coarse-grained classes to this newly introduced category. Filtering the pseudo-labels based on class ontology might work in this case but if the data is long-tailed (the new class being a head class) this might remove a lot of old data from training.

- The findings are not very surprising. If data acquisition cost is ignored and a large portion of data is labeled (which is the case in the current experimental setup) it is expected that a recent SOTA semi-supervised method will be able to reach a performance closed to supervised upper-bound accuracy trained with all data.

---

> ### Author Response · Authors · 2022-08-01
> **Rebuttal to Reviewer 6YbD (1/2)**
>
> We thank Reviewer 6YbD for the insightful comments. Reviewer 6YbD appreciates our new problem LECO, our simple and intuitive solution, and our extensive and systematic experiments. Along with the rebuttal, **we have updated our paper by including the experiments on the large-scale Mapillary dataset**, which was included in the supplement. The Mapillary dataset was collected for semantic segmentation research in the context of autonomous driving; its classes follow a long-tailed class distribution. Importantly, it reflects a **real-world case of LECO** because its ontologies were versioned from V1.2 to V2.0, formulated as TP$^0$ and TP$^1$ in our experiment. Below, we address specific issues with referred line numbers in the updated paper.
>
> > **Reviewer 6YbD thinks that (1) the proposed problem LECO is similar to continual learning, and (2) the proposed solution is an ensemble of existing techniques for other problems, which lacks technical novelty.**
>
> LECO and continual learning are two different problems (detailed in Line39-54) although both require learning models for new classes in distinct time periods (TPs). For example, LECO emphasizes the difficulty of learning for new classes which are subclasses of previously learned (super)classes (Line42). LECO formulates an underexplored but common real-world scenario as demonstrated by contemporary dataset versioning (Line35-38). In contrast, CL emphasizes catastrophic forgetting (Line93) and assumes old and new classes have clear class boundaries (Line90). Therefore, LECO is a novel problem that is quite different from CL.
>
> As a new problem, LECO offers a new testbed to exploit existing techniques developed for other problems (including class-incremental learning, fine-grained classification with both coarse and fine supervision, semi-supervised learning, etc.). We believe leveraging their insights is crucial to developing meaningful solutions to LECO. Our extensive exploration leads to novel technical solutions, i.e., one should always label new data (Line58-60), adopt the simple joint training which is surprisingly effective and can deal with inconsistent coarse-to-fine labels (Line62), use semi-supervised learning techniques to generate pseudo labels on old data and use old labels to reconcile inconsistent pseudo labels (Line65), and finetune previous TP's model which significantly outperforms training from scratch (Line69). Reviewer YxYF thinks we "combine existing methods in a *novel* way".
>
> > **Reviewer 6YbD thinks the problem setup "is restricted" because it (1) "assumes a negligible cost for data acquisition", (2) expects finer labels but not novel ones (which do not have a parent/coarse label, (3) does not consider long-tail distribution of classes.**
>
> The problem LECO is well demonstrated by dataset versioning (Line35-38), e.g., Mapillary has updated ontologies from its V1.2 to V2.0. We also repurpose the Mapillary dataset to formulate a realistic setup to study LECO, i.e., using the ontologies of its two versions in two TPs. Therefore, we argue our problem setup is quite realistic rather than restricted. We answer the three points below.
>
> 1. As explained in Line330, although data acquisition has a cost, many applications acquire data continuously regardless of the cost, e.g., acquiring data from autonomous-vehicle fleets, medical and bio-images from microscopic scanners, video frames from surveillance cameras, etc. Therefore, LECO does apply to a broad range of applications.
> 2. Our work *does* discuss the scenario when new classes emerge that have no parents (Line43). In this case, new classes can be thought of as fine-grained classes of a catch-all ${\tt background}$ class. In fact, Mapillary has such a ${\tt background}$ (aka ${\tt void}$) class in its V1.2 version, which is split into multiple new classes in V2.0, such as ${\tt temporary-barrier}$, ${\tt traffic-island}$ and ${\tt void-ground}$ (Line139). Our new experiments on Mapillary demonstrate that our solutions generalize to such real world scenarios. We added a remark on this point in Line153-159.
> 3. Our work *does* consider the long-tail distribution using real-world datasets iNaturalist and Mapillary (Line129). Long-tail distributions of classes in these datasets are depicted in the supplement (Figure 3 and 4 in supplement). Our solutions work well in the long-tailed scenario (Table 3 and 4 in main paper).

---

> > ### Author Response · Authors · 2022-08-01
> > **Rebuttal to Reviewer 6YbD (2/2)**
> >
> > > **Reviewer 6YbD comments that "the findings are not very surprising", and thinks "it is expected that a recent SOTA semi-supervised learning (SSL) method will be able to reach a performance close to supervised upper-bound accuracy trained with all data".**
> >
> > Among many findings in our work, some are quite surprising. For example, Table 2 shows that finetuning the previous TP's model significantly outperforms training-from-scratch on the same data and labels (73.64% vs. 65.69% on iNat-LECO, 30.39% vs. 27.24% on Mapillary-LECO). We conjecture that the coarse-to-fine label evolution serves as a good learning curriculum that improves performance (Line332). Moreover, while prior work shows that exploiting coarse-fine label relationships helps learning, we find that almost all the improvement can be explained by simple joint training (that *doesn’t* need one to define relationships between the old and new labels, making it applicable to far more complex ontology evolutions); cf. 82.98% vs. 83.61% on iNat-LECO in Table 3. We believe our (surprising) findings are valuable assets to our community.
> >
> > > **Reviewer 6YbD asks how the levels were selected for the iNaturalist-LECO experimental setup?**
> >
> > Good question! The selection is based on two thoughts aiming to have a clean and challenging setup to study LECO. First, we think it is good to have more fine-grained classes in TP1, hence we choose the most fine-grained level "species" in TP1. Second, we think all coarse-classes should be split later to emphasize the difficulty of learning with class-evolution, and TP0's classification task should be challenging as well with more classes. Therefore, we choose the "order" level which has 123 taxa. As reference, the original iNat dataset has seven levels: kingdom (3 taxa), phylum (8), class (29), order (123), family (339), genus (729), and species (810).
> >
> > > **Reviewer 6YbD comments that using a large validation set for hyperparameter search is not realistic, and asks whether we searched hyperparmeters for the upper-bound model.**
> >
> > Line141 explains that we sample 20% data from the training data as the validation set. This 8:2 train-val ratio is a quite common practice in the community. In real-world applications, practitioners use a validation set large enough to reliably tune hyperparameters, and then train a final model over combined train and val sets. Therefore, using 20% labeled data as the validation sets should not be an issue in our work.
> >
> > Yes, we search hyperparameters in the same way on the same validation set for the upper-bound methods.
> >
> > > **Reviewer 6YbD points out that the long-tailed class distribution might cause issues for the pseudo-labeling method, because data from tail-classes would get incorrect pseudo-labels as head-classes.**
> >
> > Great point! We verify this issue from Table 3's results on the iNat-LECO, in which both coarse and fine classes follow long-tail distributions -- the pseudo-labeling method does not work well in this long-tailed scenario. In particular, it underperforms simple supervised learning, cf. 73.56% vs. 73.64% on iNat-LECO in Table 3. However, our solutions effectively address this issue. For example, ST-Soft (self-training with soft labels) that exploits the softmax scores of pseudo-labels improves to 79.64%, combining the simple joint training boosts to 83.61%.

---

> > > ### Comment · Reviewer_6YbD · 2022-08-09
> > > **Post Rebuttal Response**
> > >
> > > I have read the rebuttal and the reviews from other reviewers. I really appreciate the explanations and additional experimental results (Mapillary dataset) provided in the rebuttal. The authors' response has adequately addressed some of my concerns. However, I am still not convinced about the novelty and significance of the conclusions. Having said that, the new results do provide additional support for the proposed solution and the promised results with multiple TPs should also be interesting. Therefore, I have decided to upgrade my score.

---

> > > > ### Author Response · Authors · 2022-08-09
> > > > **Thank you for your interaction, feedback, and upgraded score!**
> > > >
> > > > Thank you Reviewer 6YbD for your interaction, feedback, and upgraded score!
> > > >
> > > > To better understand and solve the new problem LECO, we believe it is important to rigorously leverage insights from existing approaches to related problems. We agree that doing so might overshadow the significance of our work, even though the new problem LECO formulates a broad range of applications. To strengthen our current work, as promised, we will add the new experiment in the camera-ready (cf. setup below for reference). Furthermore, we expect future work to develop novel algorithms tailored to LECO, e.g., by introducing efficient finetuning techniques across TPs, by focusing on rare fine-grained classes emerging in later TPs, etc.
> > > >
> > > > *Setup for the new experiment with more TPs.* We repurpose iNaturalist to define four TPs, with each having an ontology out of four at distinct taxa levels ["order", "family", "genus", "species"]. We will use our open-source codebase (cf. supplemental material) to readily complete this experiment.

---

### Meta-Review · Area_Chair_3Q2z · 2022-08-26

**Recommendation:** Accept
**Confidence:** Certain

**Metareview:**

The setting of evolving and refining classes over time is certainly a practical one in domains such as text classification. This paper offers some insights on questions like whether the entire data should be relabeled, or can one achieve near optimal performance by labeling only the new chunk. The paper concludes that joint training on old and new data, even if inconsistent, in conjunction with semi-supervised learning can be fairly effective.

**Award:**

No

---

### Decision · Program_Chairs · 2022-09-14

Accept